ecology

community, functional trait, limiting similarity, niche, trait hierarchy

**Author for correspondence:**
Mark K. L. Wong
e-mail: markwong.research@outlook.com

# Trait-mediated competition drives an ant invasion and alters functional diversity

Mark K. L. Wong[1,2], Roger H. Lee[3], Chi-Man Leong[3], Owen T. Lewis[1] and Benoit Guénard[3]

[1]Department of Zoology, University of Oxford, South Parks Road, Oxford OX1 3PS, UK
[2]School of Biological Sciences, The University of Western Australia, Crawley, WA 6009, Australia
[3]School of Biological Sciences, The University of Hong Kong, Kadoorie Biological Sciences Building, Pok Fu Lam Road, Hong Kong SAR, People's Republic of China

MKLW, 0000-0002-6248-3103; RHL, 0000-0002-8023-3490; C-ML, 0000-0002-6446-6518; OTL, 0000-0001-7935-6111; BG, 0000-0002-7144-1175

The assumption that differences in species' traits reflect their different niches has long influenced how ecologists infer processes from assemblage patterns. For instance, many assess the importance of environmental filtering versus classical limiting-similarity competition in driving biological invasions by examining whether invaders' traits are similar or dissimilar to those of residents, respectively. However, mounting evidence suggests that hierarchical differences between species' trait values can distinguish their competitive abilities (e.g. for the same resource) instead of their niches. Whether such trait-mediated hierarchical competition explains invasions and structures assemblages is less explored. We integrate morphological, dietary, physiological and behavioural trait analyses to test whether environmental filtering, limiting-similarity competition or hierarchical competition explain invasions by fire ants on ant assemblages. We detect both competition mechanisms; invasion success is not only explained by limiting similarity in body size and thermal tolerance (presumably allowing the invader to exploit different niches from residents), but also by the invader's superior position in trait hierarchies reflecting competition for common trophic resources. We find that the two mechanisms generate complex assemblage-level functional diversity patterns—overdispersion in some traits, clustering in others—suggesting their effects are likely missed by analyses restricted to a few traits and composite trait diversity measures.

## 1. Introduction

Invasions by alien species are a dominant threat to ecosystems and human wellbeing [1]. The mechanisms facilitating many invasions are poorly known, making it hard to predict and manage their impacts on biodiversity and ecosystem functions [2]. Still, as alien species that have invaded ecological communities are essentially members of those communities, their entry and persistence must be shaped—and may be explained—by the processes determining community composition and structure. Community assembly, therefore, provides a powerful lens through which the drivers and consequences of invasions may be illuminated [2,3].

Ecological communities are formed by the arrival of species and their subsequent interactions with abiotic and biotic environments, under the influence of processes such as dispersal, environmental filtering and biotic filtering (e.g. competition and other interspecific interactions) [4] (but see [5]). Invasion mechanisms can be viewed as manifestations of these assembly processes [2,3]. Environmental filtering determines the entry of alien species into a community in the same way as it does with native species—by selecting for species that can tolerate abiotic conditions of the local environment and excluding other species

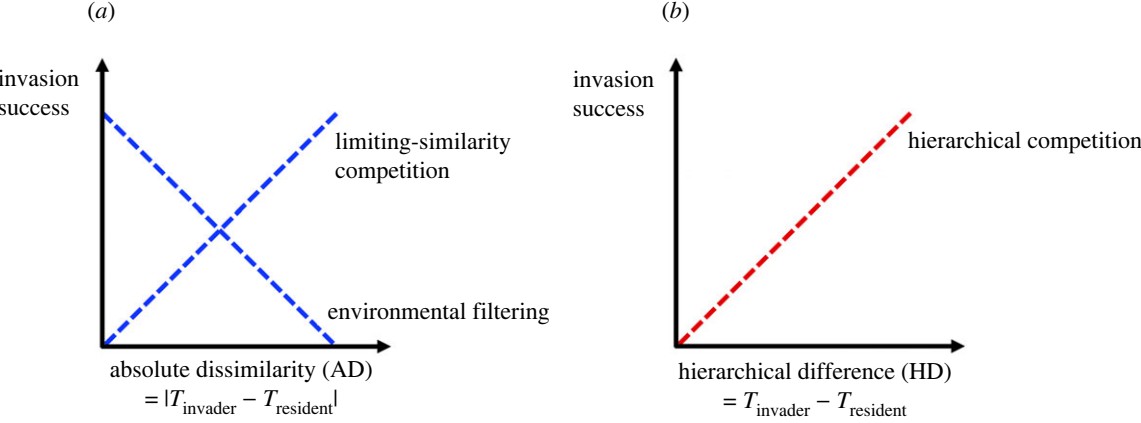

**Figure 1.** Different mechanisms involved in an invasion at the assemblage scale, and the expected relationships between invasion success and the invader's trait differences with resident species in terms of absolute dissimilarity (AD) or hierarchical difference (HD); formulae for these terms are presented below the *x*-axes ('T' refers to trait value). (*a*) If differences in a particular trait confer niche differences, and the invasion is driven by limiting-similarity competition such that the invader exploits an unused niche in the environment, invasion success will increase with increasing AD. By contrast, environmental filtering selects for an invader using similar niches as the residents, and predicts that invasion success will decrease with increasing AD. (*b*) If differences in a particular trait confer differences in competitive ability, and invasion success is determined by the invader's superior position in a competitive hierarchy which allows it to exclude weaker residents, then invasion success increases with increasing HD. Note: figure 1*b* shows the expected relationship when larger trait values confer stronger competitive abilities in a trait hierarchy; the inverse relationship is expected when smaller trait values confer stronger competitive abilities. (Online version in colour.)

that cannot tolerate those conditions [2,6,7]. Likewise, invasions driven by the mechanisms of 'niche opportunities' or 'empty niches' involve biotic filtering, specifically through limiting-similarity competition [8,9]. Here, alien species using niches—such as resources, space or time—that are dissimilar from those of native species persist in the community by exploiting those unoccupied niches, thereby avoiding intense competition that would otherwise result in their exclusion [9].

A mechanistic understanding of community assembly and invasions can be gained by examining the traits of organisms in the community, as these modulate their abiotic and biotic interactions [2,10]. For instance, studies have sought to assess the importance of environmental filtering versus limiting-similarity competition in driving invasions by comparing the extent to which invading species' traits are similar or dissimilar to those of residents, respectively [2,3,7] (figure 1*a*). Such approaches inherently assume that absolute dissimilarities in the trait values of different species in a community reflect their use of dissimilar niches (e.g. an absolute dissimilarity of 10 cm in size between a 20 cm and another 30 cm large species indicates their consumption of different prey). However, theory and increasing empirical evidence indicates that this may not necessarily be the case [11–13].

In particular, hierarchical differences in the trait values of different species can differentiate their relatively strong or weak competitive abilities, facilitating hierarchical competition for a common resource [11–13]. For instance, the success of many invasive plants is linked to their higher values (i.e. relative to native species) in traits such as maximum height and specific leaf area (figure 1*b*), which confer them stronger competitive abilities in hierarchical competition for common resources such as light [14]. While such trait-mediated hierarchical competition is known to structure various plant communities [12,13], its influence on community assembly and invasions is less explored for other taxa.

Most studies on animal communities assume *a priori* that competitive exclusion between species is solely driven by their low niche dissimilarity, as opposed to a large difference in their competitive abilities [15–17] (but see [18]).

Accordingly, attempts to detect biotic filtering typically involve tests for high absolute dissimilarities in species' trait values only [15–17]. Such approaches overlook the potential importance of hierarchical competition—not only in driving invasions, but also structuring biodiversity in general. In fact, theory predicts that a community's diversity and distribution of trait values—or 'functional diversity'—is structured very differently by limiting-similarity competition (which drives trait overdispersion) as compared to environmental filtering or hierarchical competition (which drive trait clustering) (see [15–17]).

Studies of ecological communities commonly use trait information obtained from secondary sources such as global trait databases [19] or examine traits that are most accessible to measurement. However, secondary trait information may have scant relevance to ecological interactions in the studied system [17]. Relatively accessible and commonly measured traits (e.g. morphological traits) may also correlate weakly with others (e.g. physiological and behavioural traits) that influence organisms' ecological interactions. For instance, apart from body size, behavioural traits driving antagonistic interactions in resource competition (e.g. interspecific aggression) may influence hierarchical competition in natural or invaded systems [20].

We conducted a comprehensive trait-based assessment to elucidate the mechanisms driving an invasion by the red imported fire ant, *Solenopsis invicta*, and their impacts on the functional diversity of ground-foraging ant assemblages comprising 16 species in a tropical grassland in Hong Kong. Although *S. invicta* is among the most damaging invasive species globally [1], there remains limited knowledge on the mechanisms driving its invasions. Even within North America, where *S. invicta* invasions have been intensely studied, some studies attribute *S. invicta*'s invasion success to its ability to competitively exclude native ants from nest and food resources [21], while others contend that altered abiotic conditions under anthropogenic disturbances—which favour *S. invicta*—are directly responsible for the decline of native ants [22].

We investigated how invasion success—measured as the presence and abundance of the invader *S. invicta*—across 61 plots was explained by its trait differences with the ant species at each plot. We improved on existing studies of arthropod communities [17] by examining a diverse suite of traits spanning species' morphology (body size, head width, eye width, mandible length, scape length, pronotum width and leg length), diet (trophic position), physiology (critical thermal maximum) and behaviour (interference ability). To obtain high-quality trait information, all traits were measured as continuous variables and recorded directly from multiple individuals of each species captured in field.

We measured species' trait differences in two ways: as absolute dissimilarities to approximate potential differences in niches, and as hierarchical differences to approximate potential differences in competitive abilities (following [6,12,13]) (figure 1). For each trait, we scrutinized the relationship between invasion success and species' trait differences to test for the effect of environmental filtering, limiting-similarity competition and hierarchical competition in driving the invasion (figure 1). Finally, we examined how the mechanisms at play structured the ant assemblages by comparing uninvaded and invaded assemblages in terms of the degree of overdispersion or clustering along individual trait axes and in multidimensional trait space.

## 2. Methods

### (a) Sampling ant assemblages and environmental variables

The study was conducted in a grassland with low environmental heterogeneity and sampling was performed at fine spatial scales well within ant species' dispersal ranges. The study locations were two reserves of open grassland in Hong Kong dominated by the native grass species *Arundinella setosa*. The reserves have been protected for more than 35 years, and contain networks of exposed bunds (width 5 m or less) separating permanent ponds [23]. Pilot surveys from 2015 to 2017 recorded colonies of *S. invicta* present at high densities at multiple locations [23]. From April to September 2018, we systematically characterized the ground-foraging ant assemblages at 61 plots (each a 4 × 4 m quadrat; at least 25 m between adjacent plots) using pitfall traps followed by observations at baits. Sampling at this fine spatial scale was well-suited to detecting patterns driven by biotic interactions since *S. invicta* and most ant species in the Oriental region forage within 5 m of their nests [24,25]. For the same reasons, the minimum distance of 25 m between adjacent plots facilitated independent observations. The maximum distance between any two plots was 4 km; at this spatial scale, the effects of dispersal limitation were arguably minimized as all species disperse via flying alates that can reach far greater distances [26].

In each plot, six pitfall traps (diameter: 5.5 cm) were installed to sample the ants over a period of 48 hrs. Baits were then installed within 72 h from the retrieval of the pitfall traps. Between 1000 h and 1500 h, five bait stations were installed in each plot, each comprising a slice of chicken sausage (diameter: 20 mm; height: 2 mm) on a white plastic disc (diameter: 5 cm) flushed with the ground. The sausage bait contained trophic resources required by most ants: proteins, lipids, carbohydrates and sodium. Each baiting session lasted 40 min, during which photographs were taken with a digital camera in 5 min intervals; these were subsequently used to determine the species recruiting, their abundances and inter-specific interactions (see 'Behavioural trait: interference ability'

below). The mean ground surface temperature around each bait during the 40 min session was recorded with a thermogun. In pilot trials, the 40 min duration allowed for competitive interactions to reach unequivocal outcomes and for baits to be monopolized by individual species. After each baiting session, live workers of all species encountered were collected and used for dietary and physiological trait measurements in the laboratory (below). Specimens collected from pitfall traps were used for morphological measurements, confirming species identities and determining the occurrences of species in plots. While several species collected in pitfall traps were not recorded at baits (but not vice versa), many of these were hypogaeic. Given that a main aim of the study was to investigate competition, we focused the analyses on the pool of species recorded at baits, as this best represented the ground-foraging ant assemblage limited by common resources.

At each plot, we also estimated the percentage of ground cover by applying colour thresholding techniques in IMAGEJ [27] to digital photographs, and obtained high-resolution (30 × 30 m) estimates of mean annual temperature from local climate models [28]. As these environmental factors influence ant diversity in other invaded systems [22], we used our data to test for their effects on invasion success, species richness and the trait structures of the assemblages.

### (b) Morphological traits

We measured seven morphological traits (body size, head width, eye width, mandible length, scape length, pronotum width, leg length) on at least 10 individual workers of every species ($n = 197$). These traits are linked to ant physiology and behaviour and are hypothesised to impact species performance and fitness (electronic supplementary material).

### (c) Dietary trait: trophic position

We measured the relative trophic position of each ant species using stable isotope ratios of Nitrogen ($\delta15$ N) [29]. Live ants collected from the field were killed in a $-20°C$ freezer. We then rinsed the ants with distilled water, removed their abdomens to avoid contamination by undigested material in the gut [29], and dried the samples in an oven at $40°C$ until a constant mass was reached. Dried samples with at least five workers were transferred into an aluminium capsule weighing 0.3–1 mg (workers of larger species were first ground and homogenized using a mortar and pestle following [29]). We measured the $\delta15$ N values of each sample using a Nu Perspective Isotope Ratio Mass Spectrometer coupled to a Eurovector Elemental and reported in ‰ [29]. Mineral soil collected from the field was used for baseline calibration. The $\delta15$ N value of every species was determined using 1–3 samples.

### (d) Physiological trait: critical thermal maximum

We measured the critical thermal maximum ($CT_{max}$) of individual ants following established protocols for $CT_{max}$ assays [30]. The ants were first acclimated at $25°C$ for at least 2 h in the laboratory. Individual ants were then placed in 1.5 ml Eppendorf tubes. The entrance of each tube was plugged with dry cotton to ensure that the ant was confined to an area of even temperature distribution. The tubes were then placed in a digital dry bath (BSH1004) connected to an additional thermometer (UEi Test Instruments DT302 Dual Input IP67) to ensure temperature accuracy. The assay began at a starting temperature of $36°C$, and the temperature was increased at a constant rate of $1°C\ min^{-1}$ [30]. Every 1 min, each tube was rotated and visually inspected to determine whether the ant had lost muscle coordination [30]; the temperature at which this occurred was recorded as the individual's $CT_{max}$. We measured the $CT_{max}$ of at least 10 individual workers of each species ($n = 193$).

*Proc. R. Soc. B* **289**: 20220504

**Table 1.** Results of trait and environment models for the abundance of the invader *S. invicta* across 61 plots, with standardized coefficients. Trait models explain *S. invicta* abundance as a function of its average trait differences with the resident ant species in each plot. Trait differences between *S. invicta* and resident species were measured in terms of absolute dissimilarity and hierarchical difference. Environment models were built for the percentage ground cover and mean annual temperature at each plot; these variables were also included as covariates in trait models if they improved model performance.

| model type | model name | AICc | $R^2$ | terms | $\beta$ | $p$ |
|---|---|---|---|---|---|---|
| trait | | | | | | |
| | body size | 411.7 | 0.47 | absolute dissimilarity | 0.85 | <0.05* |
| | head width | 401.6 | 0.55 | hierarchical difference | 4.5 | <0.001*** |
| | | | | ground cover | 1.77 | <0.05* |
| | $CT_{max}$ | 398.1 | 0.49 | hierarchical difference | −3.32 | <0.001*** |
| | | | | absolute dissimilarity | 3.02 | <0.001*** |
| | trophic position | 409.6 | 0.47 | hierarchical difference | 0.84 | <0.05* |
| environment | | | | | | |
| | ground cover | 428.5 | 0.0004 | ground cover | 0.2 | 0.79 |
| | temperature | 428.2 | 0.003 | temperature | 0.51 | 0.54 |

## (e) Behavioural trait: interference ability

We assessed the interference ability of ant species from observations of interspecific interactions at baits after [31] (therein termed 'behavioural dominance'). Interference ability describes a species' relative success in two types of antagonistic interactions: usurping a resource from heterospecifics (expulsion) and defending an occupied resource from usurping heterospecifics (retention) [31]. We recorded the outcomes (win/loss) in expulsion and retention incidents between ant species at baits. Each species's interference ability was then scored using the Colley rating method [31], which adjusted the value of each win and loss by the interference ability of the competitors that a species interacted with.

## (f) Data analysis
### (i) Trait selection

For each morphological trait (except body size), we corrected for the effects of body size by regressing the trait against body size and using the residuals as the new values for the trait. We then used correlation analysis and principal components analysis (PCA) to select a suite of traits that captured most interspecific variation in multidimensional trait space while minimizing redundancy from trait collinearity. Specifically, among traits showing strong positive correlations ($r > 0.7$) with one another, we selected those with stronger loadings on principal components (electronic supplementary material). Our final set of traits comprised four morphological, one dietary, one physiological and one behavioural trait: body size, head width, eye width, leg length, trophic position, $CT_{max}$ and interference ability.

### (ii) Quantifying trait differences

For each trait, we quantified two measures of trait differences between the trait value of *S. invicta* ($T_{invader}$) and the trait value of every resident ant species ($T_{resident}$) recorded at baits in the study. We calculated absolute dissimilarity as $|T_{invader} - T_{resident}|$, a non-directional measure that can indicate the magnitude of niche differences between species. We calculated hierarchical difference as $T_{invader} - T_{resident}$, a directional measure that detects differences in competitive abilities along a competitive hierarchy [6,12,13]. We then calculated the average absolute dissimilarity and hierarchical difference at each plot and used this for subsequent analysis [2,6].

### (iii) Modelling invasion success as a function of trait differences

The total number of *S. invicta* workers collected across the six pitfall traps at each of the 61 plots was used as the response term 'invader abundance', an indication of invasion success. For each trait, we modelled invader abundance as a function of the average trait difference between *S. invicta* and the assemblage of ant species in each plot. Specifically, we built a full generalized linear mixed-effects model (GLMM) with a Poisson error distribution, which included absolute dissimilarity, hierarchical difference, the environmental covariates ground cover and temperature, and an observation-level random effect to address the overdispersion in invader abundance (after [32]). We then selected the best model using a backward-stepwise variable selection procedure based on the Akaike information criterion corrected for small sample size (AICc). Prior to running the GLMMs, checks for multicollinearity were performed using correlation matrices and the 'vif' function of the car package in R [33]. Because the invader *S. invicta* had the highest value for interference ability, absolute dissimilarity equalled hierarchical difference for this trait (i.e. $r = 1$), making it theoretically impossible to separate their effects on invasion success; invasion success was thus not modelled on differences in this trait.

### (iv) Comparing patterns of trait dispersion between uninvaded and invaded assemblages

To determine whether invaded assemblages were more overdispersed or clustered relative to the uninvaded assemblages, we assessed the functional dispersion (FDisp) in each trait as well as in multidimensional trait space at the plot level (excluding *S. invicta*). We calculated FDisp using the 'fdisp' function of the FD package in R [33] and the ant species' frequencies of occurrence across the six pitfall traps at each plot. To control for potential effects of species richness on FDisp, we compared standardized effect sizes (SES) instead of the observed values [34]. We calculated SES by comparing the observed values to values generated from 999 constrained null models randomizing the matrix of species' frequencies of occurrence using the 'independent swap' algorithm. The formula for calculating SES is

$$SES = \frac{Mean_{observed} - Mean_{null}}{Standard\ Deviation_{null}}.$$

We then compared the SES values of FDisp in invaded plots to those in uninvaded plots using *t*-tests or Wilcoxon–Mann–Whitney tests (when sample variances were unequal).

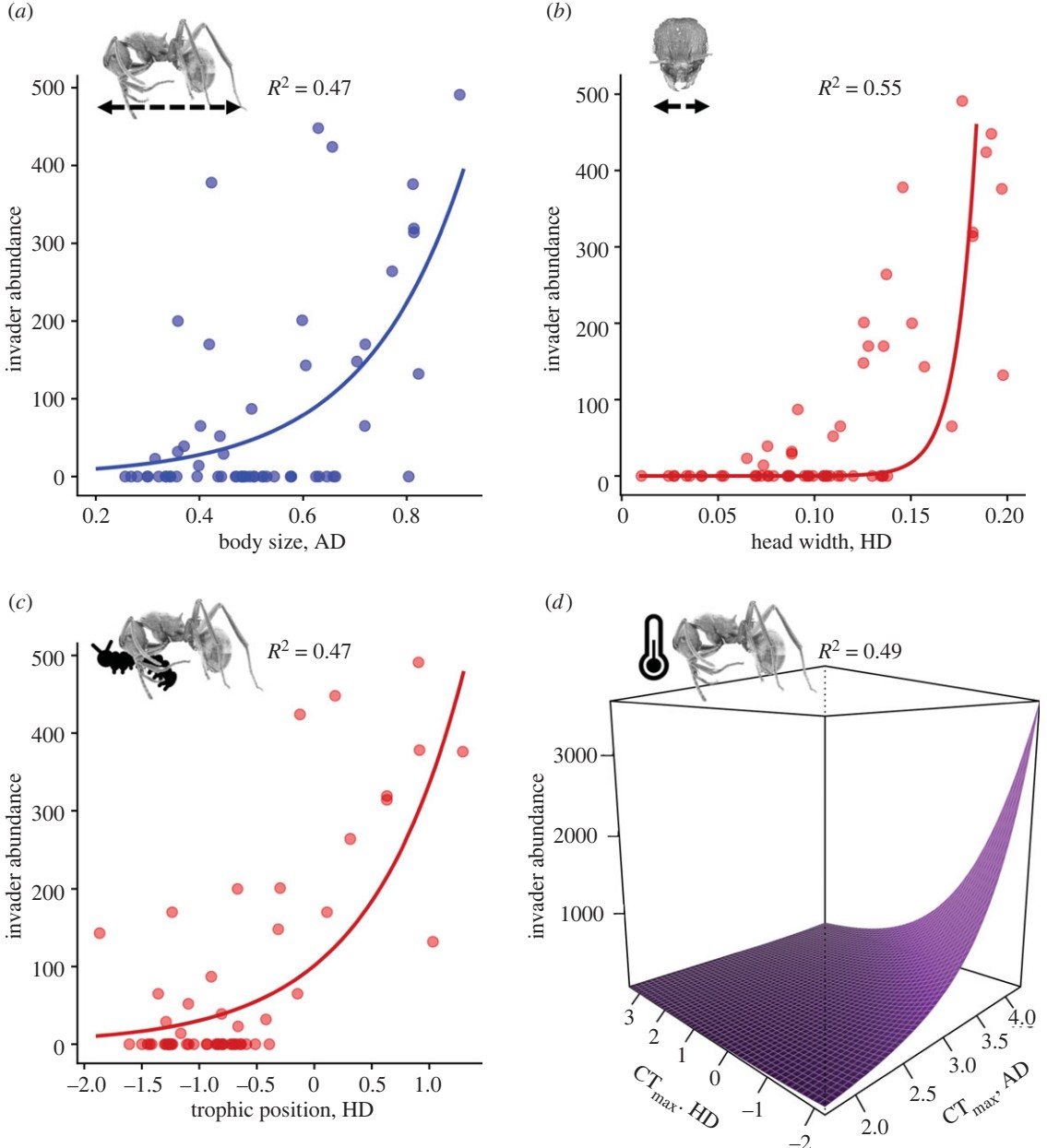

**Figure 2.** Abundance of the invader *S. invicta* as a function of its average trait differences with ant species in each of 61 plots. Two types of trait differences between *S. invicta* and ant species were measured: absolute dissimilarity (AD) and hierarchical difference (HD); their unstandardized values are presented here. The abundance of the invader increased with increasing AD in body size (*a*), as well as increasing HD in head width (*b*) and trophic position (*c*). The model for $CT_{max}$ detected significant effects of both AD and HD, and predicted higher abundances of *S. invicta* at high AD and low HD (*d*). (Online version in colour.)

## 3. Results

The abundance of the invader *S. invicta* was well explained by its average trait differences with the ant species in each plot. We detected the effects of both limiting-similarity competition and hierarchical competition in structuring the invaded assemblages, but these mechanisms were associated with separate traits (table 1 and figure 2).

Only differences in body size were clearly associated with limiting-similarity competition. The abundance of *S. invicta* was significantly higher in plots where it was dissimilar in size (i.e. either larger or smaller) than resident species (figure 2*a* and table 1). On the other hand, differences in trophic position and head width were strongly associated with hierarchical competition. The abundance of *S. invicta* was significantly higher in plots where it had a more carnivorous diet and a relatively wider head than resident species; the

latter relationship was also positively correlated with ground cover (table 1 and figure 2*b,c*). Interestingly, differences in $CT_{max}$ were associated with both forms of competition. That is, the abundance of *S. invicta* was significantly higher at plots where it had, on average, a lower $CT_{max}$ value (by as much as 2°C) but also a high absolute dissimilarity (of up to 4°C) compared to resident species (table 1 and figure 2*d*).

Compared to uninvaded ant assemblages, invaded assemblages had marginally lower species richness ($M_{Uninvaded} = 6.84 \pm 1.76$, $M_{Invaded} = 5.67 \pm 2.01$, $p = 0.02$) and were significantly more clustered in multidimensional trait space (figure 3*a*). Importantly, limiting-similarity and hierarchical competition structured the invaded assemblages along separate trait axes, with contrasting effects. Consistent with hierarchical competition, resident species in the invaded assemblages were significantly clustered in head width relative to those in the uninvaded assemblages (figure 3*b*).

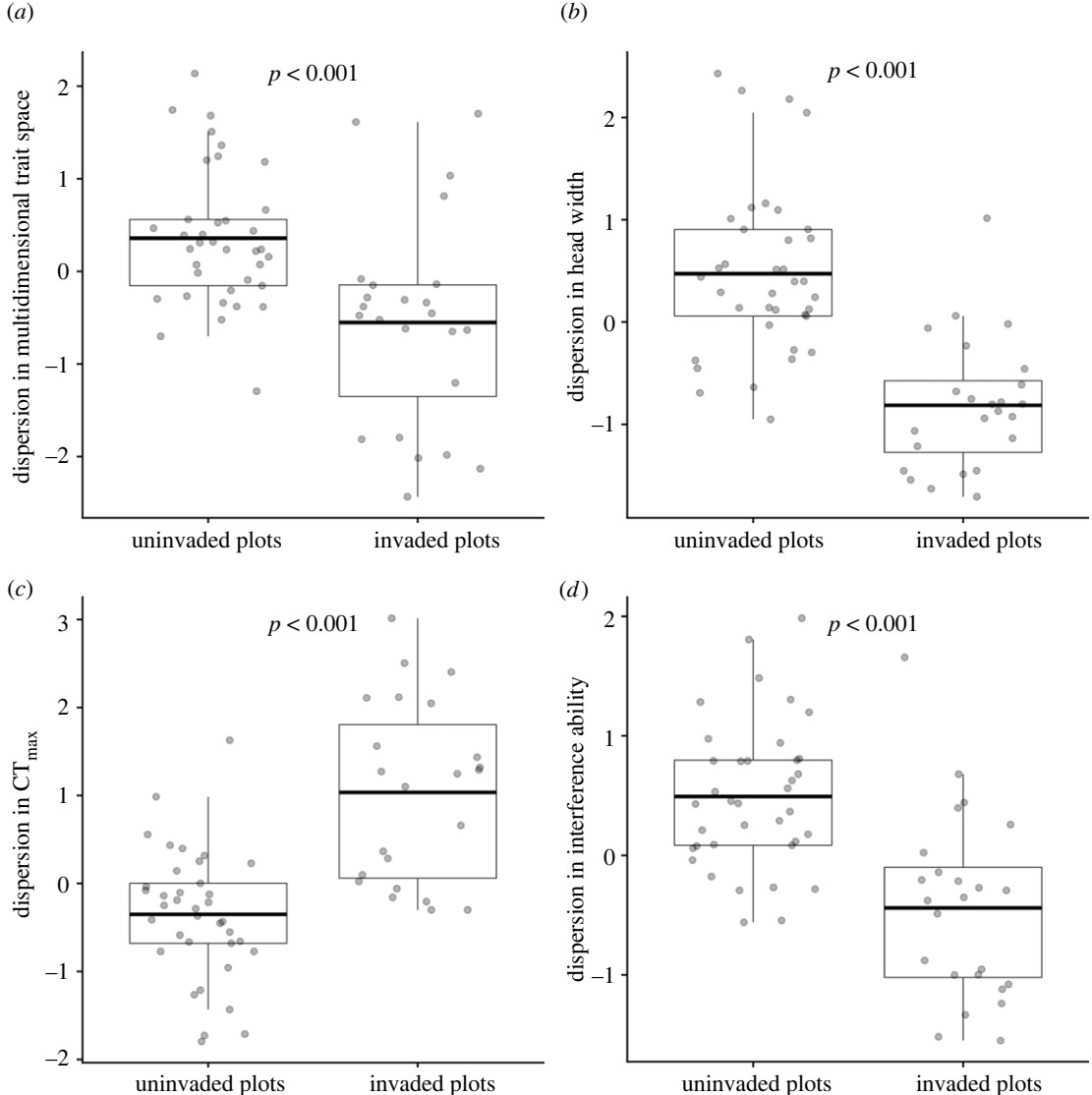

**Figure 3.** Standardized effect sizes (SES) for functional dispersion in multidimensional trait space (*a*) and three individual traits—head width (*b*), critical thermal maximum (CT$_{max}$) (*c*) and interference ability (*d*)—across 37 uninvaded and 24 invaded ant assemblages. In comparison with the uninvaded assemblages, ants in the invaded assemblages were significantly more overdispersed in CT$_{max}$ but significantly more clustered in multidimensional trait space, head width and interference ability. *p*-values correspond to results from t-tests (*b,c*) and Wilcoxon–Mann–Whitney tests (*a,d*). The invader *S. invicta* is excluded from the invaded assemblages in these analyses.

Significant clustering in interference ability was also observed (figure 3*d*). However, consistent with the effects of limiting-similarity competition, species in the invaded assemblages were more overdispersed in CT$_{max}$ than those in the uninvaded assemblages (figure 3*c*).

There was little evidence to suggest that environmental effects alone influenced invasion outcomes. The measures of absolute dissimilarity that best predicted invasion success showed positive (instead of negative) effects on invasion success (table 1), which was inconsistent with environmental filtering (figure 1). In addition, temperature and ground cover were poor predictors of invasion success (table 1) as well as the functional structures of the assemblages (electronic supplementary material).

## 4. Discussion

Here, we detected two distinct trait-mediated mechanisms driving an ant invasion: limiting-similarity competition, in

which the invader exploited different niches from residents [8,35], and hierarchical competition, in which the invader's superior trait value promoted competitive advantages over residents in competition for shared resources [11–13]. Additionally, because the two mechanisms acted distinctly on separate traits, we found contrasting patterns of overdispersion and clustering among those traits in the invaded assemblages. Our findings highlight the role of traits in conferring competitive advantages (i.e. in addition to niche opportunities) to invasive species, and the importance of trait-mediated hierarchical competition in structuring animal communities. More broadly, the findings illustrate how multiple, even opposing assembly processes may simultaneously structure functional diversity.

Consistent with the effects of limiting-similarity competition, invasion success was driven by the invader *S. invicta*'s dissimilar body size from resident ant species, which allowed it to avoid intense competition with residents by using a different niche. Notably, high similarity in body size has shown to be a good proxy of niche overlap and

a strong predictor of intense antagonistic interactions between tropical ant species. In one experiment, workers of colonies residing in mesocosms most frequently repelled (and even killed) heterospecific intruders that were the most similar to them in body size, but tolerated both smaller- and larger sized species [36]. Being substantially smaller or larger than resident species may have therefore allowed *S. invicta* to avoid such antagonistic interactions or the intense competition for body size-related resources in the grassland (e.g. types and locations of nests and foraging paths). Niche opportunities afforded by size-dissimilarities with resident species have been proposed to drive invasions by other taxa besides ants (e.g. corixids [37] and birds [38]).

Nonetheless, multiple lines of evidence suggest that trait-mediated hierarchical competition also strongly determined the invasion success of *S. invicta*. Strong associations between invasion success and *S. invicta*'s wider head and higher trophic position relative to resident ant species (table 1 and figure 2) were indicative of a competitive hierarchy for limited protein resources in the grassland. As the heads of ant workers contain the key musculature involved in the tearing and manipulation of food items [39], *S. invicta*'s relatively wider head may have allowed it to exploit trophic resources more efficiently. Notably, exploitative competition for resources among ant species can be especially intense in environments with low habitat complexity [40], such as grasslands. The relatively higher trophic position of *S. invicta* was consistent with previous observations of this species significantly depleting arthropod prey populations during invasion [21]. Additionally, significant clustering in interference ability was observed across the invaded assemblages (figure 3*d*). This pattern was consistent with the effects of hierarchical competition for a common trophic resource [11,12,16] with the invader *S. invicta* at the top of the hierarchy; it not only had the highest interference ability but also monopolized 72% of the baits in plots where it was present (electronic supplementary material). Contrary to the widely held assumption that species' trait differences mainly reflect niche differences [8,15,16], these findings suggest that differences between the invader and resident species in multiple traits were associated with differences in competitive abilities that strongly determined invasion success.

It has been proposed that interspecific trade-offs in stress tolerance and competitive dominance akin to those in plants [41] may likewise facilitate coexistence in ant assemblages [42]. Given that *S. invicta* was the top species in the competitive hierarchy for trophic resources during the daily baiting period in this study, and that it had a dissimilar or lower thermal tolerance ($CT_{max}$) compared to co-occurring resident species (figure 2*d*), it is possible that resident species exploited different thermal periods from the invader. Notably, ant species with higher thermal tolerances but lower interference ability, relatively narrower heads, and a lower trophic position than *S. invicta* were observed recruiting to baits abandoned by the invader when those baits were heated by sun exposure following changes in cloud cover (electronic supplementary material). Such dominance–tolerance trade-offs could represent equalizing mechanisms which, in addition to stabilizing mechanisms (i.e. niche differences), promoted coexistence between the invader and other ant species by mitigating the fitness impacts of interspecific differences in their competitive abilities [43].

Of prime relevance to observational studies, our findings also show that inferences about ecological processes from patterns in the multidimensional trait structure of communities may underplay the effects of distinct assembly processes that act simultaneously on different traits. Here, important effects of limiting-similarity competition in body size (figure 2*a*) and thermal tolerances (figure 2*d*) in structuring the invaded assemblages were not well reflected in their strongly clustered multidimensional trait space (figure 3*a*). This was likely due to the overwhelming, opposing effects of hierarchical competition along other trait axes such as head width (figure 3*b*) and possibly interference ability (figure 3*d*). Such effects from multiple interactive and simultaneous assembly processes on functional structure are likely underestimated, especially where biotic interactions are concerned (but see [44]).

Overall, our findings shed light on potential trait-mediated stabilizing and equalizing mechanisms that shape competition and structure invaded ant assemblages. Rigorous tests for these coexistence mechanisms would entail competition experiments measuring demographic parameters such as invasion growth rates (e.g. [45]). However, such approaches are not readily transferable to animals (see [18]). Still, we call for studies of animal communities to consider a more nuanced view of community assembly, such as by exploring how multiple assembly processes may determine community structure through their distinct effects on separate traits. In particular, the potential for both forms of trait-mediated competition (i.e. limiting-similarity and hierarchical competition) to influence community assembly should not be overlooked. Practically, this means not ruling out the role of competition and biotic filtering in community assembly simply when trait clustering is observed (as argued in [15,16]). Another interesting avenue will be to investigate the extent to which interspecific differences in certain traits consistently distinguish species' niches or competitive abilities within and across animal taxa. For instance, whereas dissimilarities in body size reflected niche differences between ant species in the present study and previous work [36], differences in the body sizes of fish species have shown to distinguish their relative competitive abilities and structure hierarchical competition [46].

Data accessibility. All data used in this study are available from the Dryad Digital Repository [47].

Electronic supplementary material is available online [48].

Authors' contributions. M.K.L.W.: conceptualization, data curation, formal analysis, funding acquisition, investigation, methodology, project administration, resources, software, validation, visualization, writing—original draft, writing—review and editing; R.H.L.: resources, writing—review and editing; C.-M.L.: resources, writing—review and editing; O.T.L.: supervision, writing—review and editing; B.G.: funding acquisition, resources, supervision, writing—review and editing.

All authors gave final approval for publication and agreed to be held accountable for the work performed therein.

Conflict of interest declaration. We declare we have no competing interests.

Funding. This study was supported by a National Geographic Grant (60-16), Clarendon Scholarship, and Forrest Research Foundation Fellowship awarded to M.K.L.W., as well as an Early Career Scheme 2017/2018 grant no. (27106417) and General Research Fund 2019/2020 grant no. (17117020) awarded from the Hong Kong Research Grants Council to B.G., R.H.L. and C.M.L.

Acknowledgements. The authors thank Sum Leung Kit for assistance with stable isotope measurements, Kaya Jumbe for assistance with baiting data, and Francois Brassard, Toby Tsang, Catherine Parr, Michelle Jackson and two anonymous reviewers for comments on previous versions of the manuscript.

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
