## [Peer Review File · Proceedings of the Royal Society B: Biological Sciences]

Review History

RSPB-2021-1365.R0 (Original submission)

Review form: Reviewer 1

Recommendation

Major revision is needed (please make suggestions in comments)

Scientific importance: Is the manuscript an original and important contribution to its field?

Good

General interest: Is the paper of sufficient general interest?

Acceptable

Quality of the paper: Is the overall quality of the paper suitable?

Acceptable

Is the length of the paper justified?

Yes

Should the paper be seen by a specialist statistical reviewer?

No

Do you have any concerns about statistical analyses in this paper? If so, please specify them explicitly in your report.

Yes

It is a condition of publication that authors make their supporting data, code and materials available - either as supplementary material or hosted in an external repository. Please rate, if applicable, the supporting data on the following criteria.

Is it accessible?

Yes

Is it clear?

Yes

Is it adequate?

Yes

Do you have any ethical concerns with this paper?

No

Comments to the Author

The role of traits patterns in explaining patterns and trends in ecology has a long history. The dataset within the study provides some insights into the complexity of traits-based analyses but further work is required to improve the rigour of the analyses. Did the authors test for collinearity of the traits? Maybe I missed this in the text (which is very densely written and could be simplified to improve clarity). Body size and thermal tolerance could be correlated. Maybe it is captured within rge data analysis "Among traits showing strong positive correlations (Supporting information), we selected those with stronger loadings on principal components." but this needs to be more explicit. i.e. strong positive correlations with one another?

It is very difficult to follow some of the arguments supporting and refuting hypotheses. This could be simplified and maybe there is simply too many theories being tested.

The figures could be improved. Make it clear what invaded and uninvaded refers to in the figure heading

The abstract, while well-written, does not stand alone. It does not portray an adequate overview of the study because there is a lack of detail. For example, mention of interspecific phenotypic variability is interesting but clarity would be improved if some example traits relevant to the study were listed. Why do complex functional diversity patterns suggest that both trait-based niche partitioning and competitive hierarchies shape responses to invasion? What multifaceted mechanisms? I am not convinced that this abstract would be accessible to all readers - the language use is complex.

Introduction

Line 125 can competitive abilities be defined simply as better or worse?

Would be useful to define early on the term "trait values".

Line 136 "finer scales" - finer than what?

Line 153 "various traits" but then only two papers - presumably these list many traits - would be useful to expand here.

Line 158 what is meant by accessible trait?

Could include a little more information on the fire ant. Why were they selected as a model species for this study? Particularly relevant would be the caste members to give background to the sampling.

Methods

Line 197 how is simple grassland defined?

Would pitfall trapping introduce any bias in traits of the individuals captured?

Were only workers sampled?

Was 10 workers for the CTmax sufficient?

Line 280-287 = results

Line 299 are pitfall traps an adequate way of assessing abundance?

Results

Line 333 body size is suggested as a proxy for niche differences consistent with Darwin's naturalization hypothesis – is this sufficiently robust? Also should reference to Darwin's naturalization hypothesis be in the discussion rather than results?

Discussion

Line 360 opening sentence is complex and could be written more simply. Just as for the abstract the first paragraph of the discussion lacks clarity and needs reading several times to gain some understanding.

Line 373 repeats introduction. Additionally could refer to other insect studies.

Line 377 why is body size so categorically used to suggest support for Darwin's naturalization hypothesis? Species of differing body sizes could exploit the niche? Seems an oversimplification and the justification of the study is to embrace complexity and nuance. The concept of niche is very complex and undefined in this study.

Line 436-438 "The strong biotic filtering detected – via distinct mechanisms such as niche partitioning and competitive hierarchies – was likely associated with the low environmental heterogeneity in the studied system and focus on relatively fine spatial scales." – could be expanded rather than simply added at the end. What do you mean by "larger scales"? How large and why?

Review form: Reviewer 2

Recommendation

Major revision is needed (please make suggestions in comments)

Scientific importance: Is the manuscript an original and important contribution to its field?

Acceptable

General interest: Is the paper of sufficient general interest?

Acceptable

Quality of the paper: Is the overall quality of the paper suitable?

Good

Is the length of the paper justified?

Yes

Should the paper be seen by a specialist statistical reviewer?

No

Do you have any concerns about statistical analyses in this paper? If so, please specify them explicitly in your report.

No

It is a condition of publication that authors make their supporting data, code and materials available - either as supplementary material or hosted in an external repository. Please rate, if applicable, the supporting data on the following criteria.

Is it accessible?

Yes

Is it clear?

Yes

Is it adequate?

Yes

Do you have any ethical concerns with this paper?

No

Comments to the Author

This is an interesting paper and is well written. The data seem great, the analyses are largely well thought out, and the results are presented clearly. I have only one major issue which is the use of the hierarchical difference metric. I have discussed this in detail below. In short, I am sceptical as to whether is telling the authors what they think it is, and whether its potential behaviour in different community contexts (assuming the invader is always "top" of the hierarchy for each trait) may lead to spurious results. The authors may be able to bat away my confusion - great! - and better explain how this metric works in the manuscript. Otherwise, some more thought may be needed in terms of trying to distinguish between these different absolute and hierarchical mechanisms. The absolute difference metric is really intuitive and makes a lot of sense.

Line 136: Perhaps just emphasise here which traits are the most competitive? I think this would really help nail down this hierarchy idea. For example: "...in these examples, plants with the largest leaf areas and largest maximum heights competed most strongly". Or something along those lines.

Line 177: "niche filtering" As in environmental filtering resulting in trait clustering? Perhaps some parentheses here clarifying would help. Sorry, but as you probably know this field is full of redundant jargon!

Figure 1: Maybe this gets cleared up later, but I'm reading it here as it is referred to in the intro. Figure 1b assumes that the invader has the most competitive trait and that it is always larger than the other traits available. Am I interpreting that correctly? Does the analysis not pick up the effect when the invader has a small value (i.e. in a case when small bodies may do better). Or am I overinterpreting a conceptual figure? I guess this is a larger point about what the trait hierarchy actually is, and whether the assumption running throughout is that the invader *does* have the

most competitive trait(s).

Line 295: OK, so the assumption is that the invader has the most competitive trait? I think this needs to be explained in more detail somewhere.

Line 301: How are the absolute and hierarchical measures calculated at the assemblage level? Above they are described as being calculated between the invader and each native species. Was an average taken to get an assemblage level value? Apologies if I missed this.

Line 307-310: This is potentially a real issue and makes interpreting these different effects difficult. It also means that using these measures only works when the invader or target species does not have the largest or smallest trait value. Is this desirable behaviour? I am not sure what the hierarchical measure is supposed to be telling us, and I'm not convinced that it is behaving in the way the authors think it is - especially when put in an assemblage context.

For example, imagine that the hierarchally best trait value is 5 on a scale of 1-10. In a community where the invader has the median trait value (5), the hierarchal difference will be ~ 0 because half the species have traits >5 and half have traits <5 . In another community, where all the native species have traits <5 , the hierarchical difference will be >0 . In line with figure 1b, we'd predict that the invader would be doing the best in the second community. But this may not necessarily be the case given that we've already assumed that the "top" of the hierarchy is the trait value that the invader has (5, in this case). I am confused by what the metric is supposed to be doing, and how it plays out in different community contexts. This issue is likely to be clouded further when in reality the invader does not have the best value for a particular trait - this may lead to some odd results and erroneous interpretations. I could be completely off base here, but I think more explanation of the behaviour of this metric is needed.

Figure 3: The y axis labels are misleading as these data are not the size or value of the spaces/traits, but are the dispersion in them. Perhaps relabel to make it clearer. This is clear in the legend but should also be obvious from the label.

Line 84: What advantage does a wider head per se confer? This is just an example, but I think more detail needs to be given as to the benefit that these different traits may offer in the grassland.

Decision letter (RSPB-2021-1365.R0)

12-Aug-2021

Dear Dr Wong:

I am writing to inform you that your manuscript RSPB-2021-1365 entitled "Multiple trait-based mechanisms shape invasions and complex functional diversity patterns" has, in its current form, been rejected for publication in Proceedings B.

This action has been taken on the advice of referees, who have recommended that substantial revisions are necessary. With this in mind we would be happy to consider a resubmission, provided the comments of the referees are fully addressed. However please note that this is not a provisional acceptance.

The resubmission will be treated as a new manuscript. However, we will approach the same reviewers if they are available and it is deemed appropriate to do so by the Editor. Please note that resubmissions must be submitted within six months of the date of this email. In exceptional

circumstances, extensions may be possible if agreed with the Editorial Office. Manuscripts submitted after this date will be automatically rejected.

Sincerely,
 Professor Gary Carvalho
 mailto: proceedingsb@royalsociety.org

Associate Editor
 Comments to Author:

I agree with both referees that while the topic of the study presented is of general interest the hypotheses tested and their predictions need refining to make the arguments more accessible. Both referees point out that the text is not very accessible in place and lack detail and context in others. While I expect the authors should be able to address the referees comments, I suspect it is a substantial revision that is needed and would not recommend it for publication in Proc Roy Soc at this time.

Reviewer(s)' Comments to Author:
 Referee: 1

Comments to the Author(s)
 The role of traits patterns in explaining patterns and trends in ecology has a long history. The dataset within the study provides some insights into the complexity of traits-based analyses but further work is required to improve the rigour of the analyses. Did the authors test for collinearity of the traits? Maybe I missed this in the text (which is very densely written and could be simplified to improve clarity). Body size and thermal tolerance could be correlated. Maybe it is captured within rge data analysis "Among traits showing strong positive correlations (Supporting information), we selected those with stronger loadings on principal components." but this needs to be more explicit. i.e. strong positive correlations with one another?

It is very difficult to follow some of the arguments supporting and refuting hypotheses. This could be simplified and maybe there is simply too many theories being tested.

The figures could be improved. Make it clear what invaded and uninvaded refers to in the figure heading

The abstract, while well-written, does not stand alone. It does not portray an adequate overview of the study because there is a lack of detail. For example, mention of interspecific phenotypic variability is interesting but clarity would be improved if some example traits relevant to the study were listed. Why do complex functional diversity patterns suggest that both trait-based

niche partitioning and competitive hierarchies shape responses to invasion? What multifaceted mechanisms? I am not convinced that this abstract would be accessible to all readers – the language use is complex.

Introduction

Line 125 can competitive abilities be defined simply as better or worse?

Would be useful to define early on the term “trait values”.

Line 136 “finer scales” – finer than what?

Line 153 “various traits” but then only two papers – presumably these list many traits – would be useful to expand here.

Line 158 what is meant by accessible trait?

Could include a little more information on the fire ant. Why were they selected as a model species for this study? Particularly relevant would be the caste members to give background to the sampling.

Methods

Line 197 how is simple grassland defined?

Would pitfall trapping introduce any bias in traits of the individuals captured?

Were only workers sampled?

Was 10 workers for the CTmax sufficient?

Line 280-287 = results

Line 299 are pitfall traps an adequate way of assessing abundance?

Results

Line 333 body size is suggested as a proxy for niche differences consistent with Darwin’s naturalization hypothesis – is this sufficiently robust? Also should reference to Darwin’s naturalization hypothesis be in the discussion rather than results?

Discussion

Line 360 opening sentence is complex and could be written more simply. Just as for the abstract the first paragraph of the discussion lacks clarity and needs reading several times to gain some understanding.

Line 373 repeats introduction. Additionally could refer to other insect studies.

Line 377 why is body size so categorically used to suggest support for Darwin’s naturalization hypothesis? Species of differing body sizes could exploit the niche? Seems an oversimplification and the justification of the study is to embrace complexity and nuance. The concept of niche is very complex and undefined in this study.

Line 436-438 “The strong biotic filtering detected – via distinct mechanisms such as niche partitioning and competitive hierarchies – was likely associated with the low environmental heterogeneity in the studied system and focus on relatively fine spatial scales.” – could be expanded rather than simply added at the end. What do you mean by “larger scales”? How large and why?

Referee: 2

Comments to the Author(s)

This is an interesting paper and is well written. The data seem great, the analyses are largely well thought out, and the results are presented clearly. I have only one major issue which is the use of the hierarchical difference metric. I have discussed this in detail below. In short, I am sceptical as to whether is telling the authors what they think it is, and whether its potential behaviour in different community contexts (assuming the invader is always "top" of the hierarchy for each trait) may lead to spurious results. The authors may be able to bat away my confusion - great! - and better explain how this metric works in the manuscript. Otherwise, some more thought may be needed in terms of trying to distinguish between these different absolute and hierarchical mechanisms. The absolute difference metric is really intuitive and makes a lot of sense.

Line 136: Perhaps just emphasise here which traits are the most competitive? I think this would really help nail down this hierarchy idea. For example: "...in these examples, plants with the largest leaf areas and largest maximum heights competed most strongly". Or something along those lines.

Line 177: "niche filtering" As in environmental filtering resulting in trait clustering? Perhaps some parentheses here clarifying would help. Sorry, but as you probably know this field is full of redundant jargon!

Figure 1: Maybe this gets cleared up later, but I'm reading it here as it is referred to in the intro. Figure 1b assumes that the invader has the most competitive trait and that it is always larger than the other traits available. Am I interpreting that correctly? Does the analysis not pick up the effect when the invader has a small value (i.e. in a case when small bodies may do better). Or am I overinterpreting a conceptual figure? I guess this is a larger point about what the trait hierarchy actually is, and whether the assumption running throughout is that the invader *does* have the most competitive trait(s).

Line 295: OK, so the assumption is that the invader has the most competitive trait? I think this needs to be explained in more detail somewhere.

Line 301: How are the absolute and hierarchical measures calculated at the assemblage level? Above they are described as being calculated between the invader and each native species. Was an average taken to get an assemblage level value? Apologies if I missed this.

Line 307-310: This is potentially a real issue and makes interpreting these different effects difficult. It also means that using these measures only works when the invader or target species does not have the largest or smallest trait value. Is this desirable behaviour? I am not sure what the hierarchical measure is supposed to be telling us, and I'm not convinced that it is behaving in the way the authors think it is - especially when put in an assemblage context.

For example, imagine that the hierarchally best trait value is 5 on a scale of 1-10. In a community where the invader has the median trait value (5), the hierarchal difference will be ~0 because half the species have traits >5 and half have traits <5. In another community, where all the native species have traits <5, the hierarchical difference will be >0. In line with figure 1b, we'd predict that the invader would be doing the best in the second community. But this may not necessarily be the case given that we've already assumed that the "top" of the hierarchy is the trait value that the invader has (5, in this case). I am confused by what the metric is supposed to be doing, and how it plays out in different community contexts. This issue is likely to be clouded further when in reality the invader does not have the best value for a particular trait - this may lead to some odd results and erroneous interpretations. I could be completely off base here, but I think more explanation of the behaviour of this metric is needed.

Figure 3: The y axis labels are misleading as these data are not the size or value of the spaces/traits, but are the dispersion in them. Perhaps relabel to make it clearer. This is clear in the legend but should also be obvious from the label.

Line 84: What advantage does a wider head per se confer? This is just an example, but I think more detail needs to be given as to the benefit that these different traits may offer in the grassland.

Author's Response to Decision Letter for (RSPB-2021-1365.R0)

See Appendix A.

RSPB-2022-0504.R0

Review form: Reviewer 3

Recommendation

Accept with minor revision (please list in comments)

Scientific importance: Is the manuscript an original and important contribution to its field?

Good

General interest: Is the paper of sufficient general interest?

Good

Quality of the paper: Is the overall quality of the paper suitable?

Good

Is the length of the paper justified?

Yes

Should the paper be seen by a specialist statistical reviewer?

No

Do you have any concerns about statistical analyses in this paper? If so, please specify them explicitly in your report.

No

It is a condition of publication that authors make their supporting data, code and materials available - either as supplementary material or hosted in an external repository. Please rate, if applicable, the supporting data on the following criteria.

Is it accessible?

Yes

Is it clear?

Yes

Is it adequate?

Yes

Do you have any ethical concerns with this paper?

No

Comments to the Author

This is an interesting study and a useful addition to the literature.

I do however have a small number of points that should be addressed:

Methods, page 5, lines 198-202

The authors state that the study took place in “two reserves of open grassland”. The composition of the grassland should be described. Whether the grasslands in question are composed mainly/exclusively of native or exotic grasses will clearly influence how native and invasive ant species interact with them. Therefore, although I assume that grassland in a reserve should be fairly intact ecologically, this needs to be described and defined in more detail. If, conversely, these grasslands are mainly exotic, then this might not be the most suitable environment in which to define ants as invasive or native.

Discussion, page 9, lines 382-383

Because it follows a statement on head width, it seems as if reference (20) infers that arthropods were depleted due to this character trait in fire ants, but (20) is in fact about the impact of polygyne fire ants on native ants and arthropods due to the abundance of worker fire ants produced by colonies. This needs to be worded more clearly.

Table S1

The table should include an extra column indicating whether each of the ant species listed is native or exotic/invasive.

A point that arises on inspection of the other species present is that *S. invicta* is shown by the study to be dominant not only against native species, but also against other species that are invasive elsewhere. This is not highlighted in the text but should be, because it sheds light not only on interactions between native and invasive species, but also between invasive species. The authors should consider (1) inclusion of other exotic/invasive species in samples as an explanatory variable to see whether there is an interaction, or (2) analysing the impact of *S. invicta* on native and invasive species separately, to further test the robustness of their hypotheses.

Review form: Reviewer 4**Recommendation**

Accept with minor revision (please list in comments)

Scientific importance: Is the manuscript an original and important contribution to its field?

Good

General interest: Is the paper of sufficient general interest?

Good

Quality of the paper: Is the overall quality of the paper suitable?

Excellent

Is the length of the paper justified?

Yes

Should the paper be seen by a specialist statistical reviewer?

No

Do you have any concerns about statistical analyses in this paper? If so, please specify them explicitly in your report.

Yes

It is a condition of publication that authors make their supporting data, code and materials available - either as supplementary material or hosted in an external repository. Please rate, if applicable, the supporting data on the following criteria.

Is it accessible?

Yes

Is it clear?

Yes

Is it adequate?

Yes

Do you have any ethical concerns with this paper?

No

Comments to the Author

The manuscript addresses the relevance of trait dissimilarities to explain the abundance of an invasive ant species in a grassland field in Hong Kong. In particular, the authors assess trait differences using average absolute difference values between invasive ant traits values and the traits values of the rest of the species of the community (Absolute Dissimilarities) and the average of the pairwise differences (Hierarchical Difference). I think that disentangle the contribution of limiting-similarity competition and hierarchical competition on species abundances is a relevant question in the field of both community assembly and species' invasions. I think as well that the manuscript is now clearly written and organized, and I am happy to recommend it for publication in the journal after addressing a few minor comments:

Lines 128-130. I would recommend rewriting this sentence to clarify more the message.

Lines 161. Clarify which databases you refer (global?) within the parentheses, you can also mention some examples of frequently used datasets in other studies.

Lines 285-290. Did the authors used means or weighted means considering the abundance of other species in the community? It is highly recommendable the use of weighted means when possible. Please clarify if you used this approach.

Lines 297-300. Please rewrite and clarify. Did authors build two parallels models one with poisson and other with negative binomial error distribution and then choose between them?

What do you mean with observation-level random effect? Please clarify if you have replicates before and which is the group level. If you do not have more than one observation per group, you do not need a hierarchical model.

Lines 304. Do you mean "differences in this trait" instead of "differences in trait".

Lines 348-350. Please be more precise about the environmental variables that resulted not significant.

Lines 415-429. Regarding to the last sentence about traits that conferring different competitive abilities in some species and niche partitioning in others; is it possible to misinterpret the role of a trait and associate it with hierarchical competition in case the higher value for the trait is reflecting a different niche but not really competing better than other species because of this trait?

I mean, could it happen that ants with higher heads have as well a demographic strategy with higher growth rates, so the reason behind ant success, in fact, is not head size?

Figure 2. (Regarding the methos behind the figure). Did you check the correlation between CTmax.HD and CTmax.AD?

Decision letter (RSPB-2022-0504.R0)

04-May-2022

Dear Dr Wong:

Your manuscript has now been peer reviewed and the reviews have been assessed by an Associate Editor. The reviewers' comments (not including confidential comments to the Editor) and the comments from the Associate Editor are included at the end of this email for your reference. As you will see, the reviewers and the Editors have raised some concerns with your manuscript and we would like to invite you to revise your manuscript to address them.

When submitting your revision please upload a file under "Response to Referees" - in the "File Upload" section. This should document, point by point, how you have responded to the reviewers' and Editors' comments, and the adjustments you have made to the manuscript. We also require a copy of the revised manuscript showing track changes to be uploaded.

Research ethics:

Use of animals and field studies:

It is a condition of publication that data supporting your paper are made available either in the electronic supplementary material. Authors must complete the 'data accessibility' section in the submission system. This should list the database and accession number for all data from the article that has been made publicly available, for instance:

NB. From April 1 2013, peer reviewed articles based on research funded wholly or partly by RCUK must include, if applicable, a statement on how the underlying research materials – such as data, samples or models – can be accessed.

[http://datadryad.org/submit?journalID=RSPB&manu=\(Document not available\)](http://datadryad.org/submit?journalID=RSPB&manu=(Document not available)) which will take you to your unique entry in the Dryad repository. If you have already submitted your data to dryad you can make any necessary revisions to your dataset by following the above link.

Please include the Dryad DOI in the Data Accessibility section and reference in the paper's bibliography.

Please see our Data Sharing Policies (<https://royalsociety.org/journals/authors/author-guidelines/>).

Please submit a copy of your revised paper within three weeks. If we do not hear from you within this time your manuscript will be rejected. If you are unable to meet this deadline please let us know as soon as possible, as we may be able to grant a short extension.

Thank you for submitting your manuscript to *Proceedings B*; we look forward to receiving your revision. If you have any questions at all, please do not hesitate to get in touch.

Best wishes,
Professor Gary Carvalho
<mailto:proceedingsb@royalsociety.org>

Associate Editor

Comments to Author:

Both reviewers are happy with the changes made by the authors in this revision. There are a few more editorial suggestions by both. However, having made these adjustments the reviewers and I suggest the manuscript should be published in *Proc Roy Soc*. Well done.

Reviewer(s)' Comments to Author:

Referee: 3

Comments to the Author(s).

This is an interesting study and a useful addition to the literature.

I do however have a small number of points that should be addressed:

Methods, page 5, lines 198-202

The authors state that the study took place in “two reserves of open grassland”. The composition of the grassland should be described. Whether the grasslands in question are composed mainly/exclusively of native or exotic grasses will clearly influence how native and invasive ant species interact with them. Therefore, although I assume that grassland in a reserve should be fairly intact ecologically, this needs to be described and defined in more detail. If, conversely, these grasslands are mainly exotic, then this might not be the most suitable environment in which to define ants as invasive or native.

Discussion, page 9, lines 382-383

Because it follows a statement on head width, it seems as if reference (20) infers that arthropods were depleted due to this character trait in fire ants, but (20) is in fact about the impact of polygyne fire ants on native ants and arthropods due to the abundance of worker fire ants produced by colonies. This needs to be worded more clearly.

Table S1

The table should include an extra column indicating whether each of the ant species listed is native or exotic/invasive.

A point that arises on inspection of the other species present is that *S. invicta* is shown by the study to be dominant not only against native species, but also against other species that are invasive elsewhere. This is not highlighted in the text but should be, because it sheds light not only on interactions between native and invasive species, but also between invasive species. The authors should consider (1) inclusion of other exotic/invasive species in samples as an explanatory variable to see whether there is an interaction, or (2) analysing the impact of *S. invicta* on native and invasive species separately, to further test the robustness of their hypotheses.

Referee: 4

Comments to the Author(s).

The manuscript addresses the relevance of trait dissimilarities to explain the abundance of an invasive ant species in a grassland field in Hong Kong. In particular, the authors assess trait differences using average absolute difference values between invasive ant traits values and the traits values of the rest of the species of the community (Absolute Dissimilarities) and the average of the pairwise differences (Hierarchical Difference). I think that disentangle the contribution of limiting-similarity competition and hierarchical competition on species abundances is a relevant question in the field of both community assembly and species' invasions. I think as well that the manuscript is now clearly written and organized, and I am happy to recommend it for publication in the journal after addressing a few minor comments:

Lines 128-130. I would recommend rewriting this sentence to clarify more the message.

Lines 161. Clarify which databases you refer (global?) within the parentheses, you can also mention some examples of frequently used datasets in other studies.

Lines 285-290. Did the authors used means or weighted means considering the abundance of other species in the community? It is highly recommendable the use of weighted means when possible. Please clarify if you used this approach.

Lines 297-300. Please rewrite and clarify. Did authors build two parallels models one with poisson and other with negative binomial error distribution and then choose between them?

What do you mean with observation-level random effect? Please clarify if you have replicates before and which is the group level. If you do not have more than one observation per group, you do not need a hierarchical model.

Lines 304. Do you mean “differences in this trait” instead of “differences in trait”.

Lines 348-350. Please be more precise about the environmental variables that resulted not significant.

Lines 415-429. Regarding to the last sentence about traits that conferring different competitive abilities in some species and niche partitioning in others; is it possible to misinterpret the role of a trait and associate it with hierarchical competition in case the higher value for the trait is reflecting a different niche but not really competing better than other species because of this trait? I mean, could it happen that ants with higher heads have as well a demographic strategy with higher growth rates, so the reason behind ant success, in fact, is not head size?

Figure 2. (Regarding the methos behind the figure). Did you check the correlation between CTmax.HD and CTmax.AD?

Author's Response to Decision Letter for (RSPB-2022-0504.R0)

See Appendix B.

RSPB-2022-0504.R1 (Revision)

Review form: Reviewer 4

Recommendation

Accept as is

Scientific importance: Is the manuscript an original and important contribution to its field?

Good

General interest: Is the paper of sufficient general interest?

Good

Quality of the paper: Is the overall quality of the paper suitable?

Excellent

Is the length of the paper justified?

Yes

Should the paper be seen by a specialist statistical reviewer?

No

Do you have any concerns about statistical analyses in this paper? If so, please specify them explicitly in your report.

No

It is a condition of publication that authors make their supporting data, code and materials available - either as supplementary material or hosted in an external repository. Please rate, if applicable, the supporting data on the following criteria.

Is it accessible?

Yes

Is it clear?

N/A

Is it adequate?

Yes

Do you have any ethical concerns with this paper?

No

Comments to the Author

The authors have addressed all my concerns and done a good job improving the manuscript. I'm happy to recommend the current version for publication.

I just have a last comment (which could be directly addressed before publishing edition without responding again to reviewers) regarding to model multicollinearity test:

Last response page 6 in Authors response letter. I suggest specifying the functions and/or packages used to test variables' collinearity.

That's all. Congratulations and thanks for this interesting study.

Decision letter (RSPB-2022-0504.R1)

31-May-2022

Dear Dr Wong

I am pleased to inform you that your Review manuscript RSPB-2022-0504.R1 entitled "Trait-mediated competition drives an ant invasion and alters functional diversity" has been accepted for publication in Proceedings B.

The referee(s) do not recommend any further changes. Therefore, please proof-read your manuscript carefully and upload your final files for publication. Because the schedule for publication is very tight, it is a condition of publication that you submit the revised version of your manuscript within 7 days. If you do not think you will be able to meet this date please let me know immediately.

To upload your manuscript, log into <http://mc.manuscriptcentral.com/prsb> and enter your Author Centre, where you will find your manuscript title listed under "Manuscripts with Decisions." Under "Actions," click on "Create a Revision." Your manuscript number has been appended to denote a revision.

You will be unable to make your revisions on the originally submitted version of the manuscript. Instead, upload a new version through your Author Centre.

- 1) A text file of the manuscript (doc, txt, rtf or tex), including the references, tables (including captions) and figure captions. Please remove any tracked changes from the text before submission. PDF files are not an accepted format for the "Main Document".

2) A separate electronic file of each figure (tiff, EPS or print-quality PDF preferred). The format should be produced directly from original creation package, or original software format. Please note that PowerPoint files are not accepted.

3) Electronic supplementary material: this should be contained in a separate file from the main text and the file name should contain the author's name and journal name, e.g. `authorname_procb_ESM_figures.pdf`

All supplementary materials accompanying an accepted article will be treated as in their final form. They will be published alongside the paper on the journal website and posted on the online figshare repository. Files on figshare will be made available approximately one week before the accompanying article so that the supplementary material can be attributed a unique DOI. Please see: <https://royalsociety.org/journals/authors/author-guidelines/>

4) Data-Sharing and data citation

It is a condition of publication that data supporting your paper are made available. Data should be made available either in the electronic supplementary material or through an appropriate repository. Details of how to access data should be included in your paper. Please see <https://royalsociety.org/journals/ethics-policies/data-sharing-mining/> for more details.

<http://datadryad.org/submit?journalID=RSPB&manu=RSPB-2022-0504.R1> which will take you to your unique entry in the Dryad repository.

Once again, thank you for submitting your manuscript to Proceedings B and I look forward to receiving your final version. If you have any questions at all, please do not hesitate to get in touch.

Sincerely,
Professor Gary Carvalho
Editor, Proceedings B
proceedingsb@royalsociety.org

Referee: 4

Comments to the Author(s)

The authors have addressed all my concerns and done a good job improving the manuscript. I'm happy to recommend the current version for publication.

I just have a last comment (which could be directly addressed before publishing edition without responding again to reviewers) regarding to model multicollinearity test:

Last response page 6 in Authors response letter. I suggest specifying the functions and/or packages used to test variables' collinearity.

That's all. Congratulations and thanks for this interesting study.

Decision letter (RSPB-2022-0504.R2)

01-Jun-2022

Dear Dr Wong

I am pleased to inform you that your manuscript entitled "Trait-mediated competition drives an ant invasion and alters functional diversity" has been accepted for publication in Proceedings B.

Your article has been estimated as being 9 pages long. Our Production Office will be able to confirm the exact length at proof stage.

Data Accessibility section

Open Access

Paper charges

Sincerely,

Proceedings B

Appendix A

Associate Editor

I agree with both referees that while the topic of the study presented is of general interest the hypotheses tested and their predictions need refining to make the arguments more accessible. Both referees point out that the text is not very accessible in place and lack detail and context in others. While I expect the authors should be able to address the referees comments, I suspect it is a substantial revision that is needed and would not recommend it for publication in Proc Roy Soc at this time.

Response: We thank the editor and both referees for their interest in the work and positive assessment of the research, as well as for the opportunity to thoroughly revise and improve the writing and presentation of the manuscript.

As suggested by the editor, we have since undertaken a rigorous revision of the manuscript to address the specific comments of both referees. Briefly, this has involved a complete re-writing of the Abstract and Introduction, extensive revisions and restructuring of the text in the Discussion, as well as revisions to the Methods and Figures to improve their clarity. Despite the extensive changes made to the writing and the communication of the findings, our results remain the same, and we believe that our manuscript is substantially improved as a result.

Our detailed responses to both referees' specific comments are provided below.

Thank you for your consideration of our revised manuscript.

Referee: 1

The abstract, while well-written, does not stand alone. It does not portray an adequate overview of the study because there is a lack of detail. For example, mention of interspecific phenotypic variability is interesting but clarity would be improved if some example traits relevant to the study were listed. Why do complex functional diversity patterns suggest that both trait-based niche partitioning and competitive hierarchies shape responses to invasion? What multifaceted mechanisms? I am not convinced that this abstract would be accessible to all readers – the language use is complex.

Response: We agree with the reviewer that the previous abstract was indeed too 'abstract' and did not portray an adequate overview of our study. Accordingly, we have completely re-written the abstract to simplify the language used, and to make explicit (i) the conceptual background and knowledge gap, (ii) the three hypothetical processes we tested for, (iii) the two processes that received support in our empirical study, (iv) the specific traits that showed these relationships, and (v) the implications for other work. The new abstract is as follows:

"The assumption that differences in species' traits reflect their different niches has long influenced how ecologists infer processes from assemblage patterns. For instance, many

assess the importance of environmental filtering versus classical limiting-similarity competition in driving biological invasions by examining whether invaders' traits are similar or dissimilar to those of residents, respectively. However, mounting evidence suggests that hierarchical differences between species' trait values can distinguish their competitive abilities (e.g., for the same resource) instead of their niches. Whether such trait-mediated hierarchical competition explains invasions and structures assemblages is less explored. We integrate morphological, dietary, physiological and behavioural trait analyses to test whether environmental filtering, limiting-similarity competition, or hierarchical competition explain invasions by fire ants on ant assemblages. We detect both competition mechanisms; invasion success is not only explained by limiting similarity in body size and thermal tolerance (presumably allowing the invader to exploit different niches from residents), but also by the invader's superior position in trait hierarchies reflecting competition for common trophic resources. We find that the two mechanisms generate complex assemblage-level functional diversity patterns – overdispersion in some traits, clustering in others – suggesting their effects are likely missed by analyses restricted to a few traits and composite trait diversity measures.”

It is very difficult to follow some of the arguments supporting and refuting hypotheses. This could be simplified and maybe there is simply too many theories being tested.

Response: We agree that the text in the previous version of the manuscript (especially that in the Introduction) was too convoluted, and could therefore be inaccessible to the general readers of Proc B. Following the referee's useful recommendations, we have thoroughly re-written the Introduction, and reduced the number of hypotheses / theories that were previously discussed by synthesizing these into three hypotheses.

Our new, simpler Introduction generally reflects the structure of our revised Abstract. In the Introduction, we now clearly outline the three hypotheses (i.e., assembly processes) that our study tests in an invasion context: environmental filtering, limiting-similarity competition, and hierarchical competition. We discuss how these hypotheses predict different relationships in terms of the trait differences between species, and emphasise that most studies of animal communities (and invasions) have probably overlooked hierarchical competition. We then briefly highlight another general limitation of trait studies – the use of secondary, coarse data on traits that capture only few phenotypic dimensions (e.g., analyses that only use morphological traits) – that we aim to overcome in our study. Finally, we describe our study system (i.e., fire ant invasions) and its relevance to the hypotheses.

We hope the new and simplified Introduction suffices to provide readers with the relevant background and research goals in an accessible manner.

Did the authors test for collinearity of the traits? Maybe I missed this in the text (which is very densely written and could be simplified to improve clarity). Body size and thermal tolerance could be correlated. Maybe it is captured within the data analysis "Among traits showing strong positive correlations (Supporting information), we selected those with stronger loadings on

principal components." but this needs to be more explicit. i.e. strong positive correlations with one another?

Response: Yes, trait selection was performed specifically to avoid the problem of trait collinearity. We have incorporated the referee's suggestion to clarify this point more explicitly in the text (please refer to the revised sentences below). With regards to the referee's additional comment, body size and thermal tolerance were weakly negatively correlated in this instance ($r=-0.42$); thus, it was possible to include both in the final suite of traits used in the analyses.

"We then used correlation analysis and Principal Components Analysis (PCA) to select a suite of traits that captured most interspecific variation in multidimensional trait space while minimizing redundancy from trait collinearity. Specifically, among traits showing strong positive correlations ($r > 0.7$) with one another, we selected those with stronger loadings on principal components (Supporting information)."

The figures could be improved. Make it clear what invaded and uninvaded refers to in the figure heading.

Response: To improve the clarity of Figure 3, we have since revised the labels of the x- and y-axes, as follows:

x-axis labels: from "Uninvaded" and "Invaded" to "Uninvaded plots" and "Invaded plots"

y-axis labels: The phrase "Dispersion in" is now included before the name of each trait variable, so as to clarify the precise aspect of trait diversity that is being shown, for instance, "Dispersion in Head Width".

Introduction

Line 125 can competitive abilities be defined simply as better or worse?

Response: We wish to clarify that in writing that "*hierarchical differences in species' trait values can differentiate their (better or worse) competitive abilities in competitive hierarchies...*", we were not attempting to define competitive abilities; rather, simply illustrating that hierarchical differences in the trait values of separate species can differentiate those species in terms of their relative competitive abilities (which may be stronger for some species and weaker for other species). We have since revised the sentence for clarity.

"...hierarchical differences in the trait values of different species can differentiate their relatively strong or weak competitive abilities, facilitating hierarchical competition for a common resource"

Would be useful to define early on the term "trait values".

Response: We have since added text at the first mention of the term ‘trait values’ to clearly illustrate what these might look like.

“...absolute dissimilarities in the trait values of different species in a community reflect their use of dissimilar niches (e.g., an absolute dissimilarity of 10 cm in size between a 20-cm and another 30-cm large species indicates their consumption of different prey).”

Line 136 “finer scales” – finer than what?

Response: The sentence the referee refers to has since been removed during our thorough revision of the Introduction for clarity.

Line 153 “various traits” but then only two papers – presumably these list many traits – would be useful to expand here.

Response: This is a useful suggestion. We have since revised the sentence to illustrate examples of these traits.

“the success of many plant invaders is linked to their higher values (i.e., relative to native species) in traits such as maximum height and specific leaf area (Figure 1b), which confer them stronger competitive abilities in hierarchical competition for common resources such as light.”

Line 158 what is meant by accessible trait?

Response: We have clarified this in the revised sentence, as follows:

“Studies of ecological communities commonly use trait information obtained from secondary sources (e.g., databases) or examine traits that are most accessible to measurement.”

Could include a little more information on the fire ant. Why were they selected as a model species for this study?

Response: We have since provided more background to justify the selection of *S. invicta* as a model species for the study.

“We conducted a comprehensive trait-based assessment to elucidate the mechanisms driving an invasion by the Red Imported Fire Ant, *Solenopsis invicta*, and their impacts on the functional diversity of ground-foraging ant assemblages comprising 16 species in a tropical grassland in Hong Kong. Although *S. invicta* is among the most damaging invasive species globally (1), there remains limited knowledge on the mechanisms driving its invasions. Even within North America, where *S. invicta* invasions have been intensely studied, some studies attribute invasion success to *S. invicta*’s ability to competitively exclude native ants from nest and food resources (20), while others contend that altered abiotic conditions under anthropogenic disturbances – which favour *S. invicta* – are directly responsible for the decline of native ants (21).”

Methods

Line 197 how is simple grassland defined?

Response: We have since replaced the phrase ‘simple grassland setting’ with ‘grassland’, as the word ‘simple’ has an ambiguous meaning in this context, as indicated by the referee. The word is also redundant because the sentence subsequently states that the grassland had ‘low environmental heterogeneity’. The new sentence is:

“The study was conducted in a grassland with low environmental heterogeneity and sampling was performed at fine spatial scales well within ant species’ dispersal ranges.”

Would pitfall trapping introduce any bias in traits of the individuals captured?

Response: This is an insightful and relevant question. Indeed, our recent research (Lee & Guénard, 2020) examined this precise issue in the field, and found that forest ant assemblages sampled by pitfall traps were dominated by larger sized, slender-shaped, and long-legged species relative to those sampled by extractions of leaf-litter. Such potential biases arising from pitfall trapping are certainly relevant to studies aiming to inventory entire ant assemblages across various vertical habitat strata (i.e., the soil, litter, ground surface, and various layers of vegetation). However, this was not the focus of the present study. As stated in the Methods, “*Given that a main aim of the study was to investigate competition, we focused the analyses on the pool of species recorded at baits, as these best represented the ground-foraging ant assemblage limited by common resources.*” To this end, we deemed pitfall traps to be suitable for our study on ground-foraging ant assemblages because they have shown to effectively capture the same ground-foraging ant species recorded at baits (Wang et al., 2001; Lopes & Vasconcelos, 2008); this was also confirmed by our data, which showed that all species recorded at baits were also recorded in the pitfall traps at those sites.

Lee, R. H., & Guénard, B. (2019). Choices of sampling method bias functional components estimation and ability to discriminate assembly mechanisms. *Methods in Ecology and Evolution*, 10(6), 867-878.

Lopes, C. T., & Vasconcelos, H. L. (2008). Evaluation of three methods for sampling ground-dwelling ants in the Brazilian Cerrado. *Neotropical Entomology*, 37(4), 399-405.

Wang, C., Strazanac, J., & Butler, L. (2001). A comparison of pitfall traps with bait traps for studying leaf litter ant communities. *Journal of Economic Entomology*, 94(3), 761-765.

Were only workers sampled?

Response: Yes, only workers were sampled in this study on interspecific competition among ant species. This is a reasonable approach because workers not only form the bulk of a colony's biomass, but are essentially the functional units of the colony; the majority of interactions between a given colony and its abiotic and biotic environments (as well as the positive or negative consequences of these interactions on colony fitness) would either occur directly or indirectly through the workers. The approach of focusing on ant workers it is also consistent with almost all studies that have examined competition between ant species to date (Parr & Gibb, 2010). Nonetheless, the roles of queen ants in interspecific interactions such as competition is certainly an intriguing and potentially important (though very logistically challenging) area that would need to be addressed in future research.

Parr, C. L., & Gibb, H. (2010). Competition and the role of dominant ants. In: Lach, L., Parr, C. L. & Abbott, K. L. (Eds.): *Ant Ecology*, Oxford University Press, Oxford, pp. 77–96.

Was 10 workers for the CT_{max} sufficient?

Response: We believe that a minimum sample of 10 workers per species was sufficient for deriving a mean CT_{max} value for each species in our study. To date, most comparative studies on ant species' thermal tolerances have used comparable or lower sample sizes to estimate species' mean CT_{max} values (e.g., 5 workers, Bujan & Kaspari, 2017; Roeder et al., 2018; 7 workers, Boyle et al., 2021; 10 workers, Andrew et al., 2019). Furthermore, our recent research specifically examining intraspecific as well as interspecific variation in the CT_{max} values of ant species in Hong Kong (Leong et al., in press) further suggests that a sample of 10 workers suffices for deriving species-mean CT_{max} values for interspecific comparisons in studies of community ecology.

Andrew, N. R., Miller, C., Hall, G., Hemmings, Z., & Oliver, I. (2019). Aridity and land use negatively influence a dominant species' upper critical thermal limits. *PeerJ*, 6, e6252.

Boyle, M. J., Bishop, T. R., Luke, S. H., van Breugel, M., Evans, T. A., Pfeifer, M., ... & Ewers, R. M. (2021). Localised climate change defines ant communities in human-modified tropical landscapes. *Functional Ecology*, 35(5), 1094-1108.

Bujan, J. F., & Kaspari, M. (2017). Nutrition modifies critical thermal maximum of a dominant canopy ant. *Journal of Insect Physiology*, 102, 1-6.

Leong, C. M., Tsang, T. P., & Guénard, B. Testing the reliability and ecological implications of ramping rates in the measurement of Critical Thermal maximums. *Plos One* (in press).

Roeder, K. A., Roeder, D. V., & Kaspari, M. (2018). The role of temperature in competition and persistence of an invaded ant assemblage. *Ecological Entomology*, 43, 774-781.

Line 280-287 = results

Response: We agree with the reviewer that these descriptions of the traits' loadings on different principal components should not be included in the Methods; they would more commonly be reported as results in other studies. Nonetheless, as the principal loadings do not pertain to the main results of our study, but instead describe a preliminary step of trait selection to facilitate the main analyses, we have moved the relevant text to the Supporting information. The revised text describing trait selection in the Methods is as follows:

“We then used correlation analysis and Principal Components Analysis (PCA) to select a suite of traits that captured most interspecific variation in multidimensional trait space while minimizing redundancy from trait collinearity. Specifically, among traits showing strong positive correlations ($r > 0.7$) with one another, we selected those with stronger loadings on principal components (Supporting information). Our final set of traits comprised four morphological, one dietary, one physiological and one behavioural trait: Body Size, Head Width, Eye Width, Leg Length, Trophic Position, CTmax, and Interference Ability.”

Line 299 are pitfall traps an adequate way of assessing abundance?

Response: Yes. Pitfall traps have been very widely used to assess the abundances of ant species in numerous studies on ant assemblages (reviewed in Gibb et al., 2017), including studies on invasions of ant assemblages by *S. invicta* (e.g., Porter & Savignano, 1990; Gotelli & Arnett, 2000; Morrison, 2002; Stuble et al., 2009).

Gotelli, N. J., & Arnett, A. E. (2000). Biogeographic effects of red fire ant invasion. *Ecology Letters*, 3(4), 257-261.

Gibb, H., Dunn, R. R., Sanders, N. J., Grossman, B. F., Photakis, M., Abril, S., ... & Parr, C. L. (2017). A global database of ant species abundances. *Ecology*, 98, 883-884.

King, J. R., & Tschinkel, W. R. (2008). Experimental evidence that human impacts drive fire ant invasions and ecological change. *Proceedings of the National Academy of Sciences*, 105(51), 20339-20343.

Morrison, L. W. (2002). Long-term impacts of an arthropod-community invasion by the imported fire ant, *Solenopsis invicta*. *Ecology*, 83(8), 2337-2345.

Porter, S. D., & Savignano, D. A. (1990). Invasion of polygyne fire ants decimates native ants and disrupts arthropod community. *Ecology*, 71(6), 2095-2106.

Stuble, K. L., Kirkman, L. K., & Carroll, C. R. (2009). Patterns of abundance of fire ants and native ants in a native ecosystem. *Ecological Entomology*, 34(4), 520-526.

Results

Line 333 body size is suggested as a proxy for niche differences consistent with Darwin's naturalization hypothesis – is this sufficiently robust? Also should reference to Darwin's naturalization hypothesis be in the discussion rather than results?

Response: We have since removed the reference to Darwin's naturalization hypothesis in the Results section, as we agree with the reviewer that any references to theory should be discussed in the Discussion section. The revised sentence describing the results in terms of body size now clarifies why differences in body size between *S. invicta* and resident ant species were consistent with limiting-similarity competition.

“Only differences in Body Size were clearly associated with limiting-similarity competition. The abundance of *S. invicta* was significantly higher in plots where it was dissimilar in size (i.e., either larger or smaller) than resident species.”

Discussion

Line 360 opening sentence is complex and could be written more simply. Just as for the abstract the first paragraph of the discussion lacks clarity and needs reading several times to gain some understanding.

Response: We have deleted the previous opening sentence in the Discussion, which was complex and vague in meaning. The new opening sentence immediately describes the findings in plain and direct terms. The subsequent sentences have also been simplified for clarity.

“Here, we detected two distinct trait-mediated mechanisms driving an ant invasion: limiting-similarity competition, in which the invader exploited different niches from residents (8,35), and hierarchical competition, in which the invader's superior trait value promoted competitive advantages over residents in competition for shared resources (11-13). Additionally, because the two mechanisms acted distinctly on separate traits, we found contrasting patterns of overdispersion and clustering among those traits in the invaded assemblages. Our findings highlight the role of traits in conferring competitive advantages (i.e., in addition to niche opportunities) to invasive species, and the importance of trait-mediated hierarchical competition in structuring animal communities. More generally, the findings illustrate how multiple, even opposing assembly processes may simultaneously structure functional diversity.”

Line 373 repeats introduction. Additionally could refer to other insect studies.

Response: The topic of stabilizing and equalizing mechanisms (i.e., the subject of “Line 373” previously) is no longer discussed in the new Introduction, so the issue of repetition has been resolved. In revising the Discussion to make the text more accessible to the broad readership of Proc B, we have also minimised speculative statements about theoretical topics (e.g., stabilizing and equalizing mechanisms). The paragraph that the referee refers to has since been revised to explain how dissimilarities in body size between the invader *S. invicta* and the resident ants may

correspond to their use of different niches and allow those species to avoid intense competition. We have expanded our previous arguments and cited literature to support these claims, including a very relevant experimental study on ants, as well as observations of the invasions by other taxa. Please see the excerpt of that paragraph in our response to the comment below.

Line 377 why is body size so categorically used to suggest support for Darwin's naturalization hypothesis? Species of differing body sizes could exploit the niche?

Response: As suggested by the referee, we have completely revised the manuscript to reduce any unnecessary references to multiple theories and mechanisms. As such, we no longer discuss Darwin's naturalization hypothesis, but the broader assembly process that it manifests from – limiting-similarity competition, which assumes that trait differences between species correspond to their niche differences. In the revised Discussion, we explain why dissimilarities in body size between the invader *S. invicta* and the resident ants may correspond to their use of different niches and allow those species to avoid intense competition. We have expanded our previous arguments and cited literature to support these claims, including a very relevant experimental study on ants, as well as observations of the invasions by other taxa.

“Consistent with the effects of limiting-similarity competition, invasion success was driven by the invader *S. invicta*'s dissimilar body size from resident ant species, which allowed it to avoid intense competition with residents by using a different niche. Notably, a high similarity in body size has shown to be a good proxy of niche overlap and a strong predictor of intense, antagonistic interactions between tropical ant species. In one experiment, workers of colonies residing in mesocosms most frequently repelled (and even killed) heterospecific intruders that were the most similar to them in body size, but tolerated both smaller- and larger-sized species (36). Being substantially smaller or larger than resident species may have therefore allowed *S. invicta* to avoid such antagonistic interactions or the intense competition for body size-related resources in the grassland (e.g., types and locations of nests and foraging paths). Niche opportunities afforded by size-dissimilarities with resident species have been proposed to drive invasions by other taxa besides ants [e.g., corixids (37) and birds (38)].”

Line 436-438 “The strong biotic filtering detected – via distinct mechanisms such as niche partitioning and competitive hierarchies – was likely associated with the low environmental heterogeneity in the studied system and focus on relatively small spatial scales.” – could be expanded rather than simply added at the end.

Response: In thoroughly revising the Discussion for clarity, we have removed this statement. Our new concluding paragraph outlines what we feel are the key takeaways from the work and the most interesting directions for future research. An excerpt of this paragraph is provided below.

“Overall, our findings shed light on potential trait-mediated stabilizing and equalizing mechanisms that shape competition and structure invaded ant assemblages. Rigorous tests for these coexistence mechanisms would entail competition experiments measuring demographic

parameters such as invasion growth rates (e.g., 45). However, such approaches are not readily transferable to animals (see 15). Still, we call for studies of animal communities to consider a more nuanced view of community assembly, such as by exploring how multiple assembly processes may determine community structure through their distinct effects on separate traits. In particular, the potential for both forms of trait-mediated competition (i.e., limiting-similarity and hierarchical competition) to influence community assembly should not be overlooked. Practically, this means not ruling out the role of competition and biotic filtering in community assembly simply when trait clustering is observed (as argued by 16,17). Another interesting avenue will be to investigate the extent to which interspecific differences in certain traits consistently distinguish species' niches or relative competitive abilities within and across animal taxa. For instance, whereas dissimilarities in body size reflected niche differences between ant species in the present study and previous work (36), differences in body sizes among fish species have shown to distinguish their relative competitive abilities and structure hierarchical competition (46).”

What do you mean by “larger scales”? How large and why?

Response: In thoroughly revising the Discussion for clarity, we have removed this statement. Our new concluding paragraph outlines what we feel are the key takeaways from the work and the most interesting directions for future research. An excerpt of this paragraph is provided in our response to the comment above.

Referee: 2

This is an interesting paper and is well written. The data seem great, the analyses are largely well thought out, and the results are presented clearly.

Response: We thank the referee for their positive assessment of the manuscript.

Line 136: Perhaps just emphasise here which traits are the most competitive? I think this would really help nail down this hierarchy idea. For example: "...in these examples, plants with the largest leaf areas and largest maximum heights competed most strongly". Or something along those lines.

Response: Thank you for this useful suggestion. While the sentence that the referee refers to has since been removed during our re-writing of the Introduction, we have applied the referee's suggestion to emphasise the more competitive trait values in another example. A citation to a highly relevant paper that demonstrates and explains such relationships is also provided:

“For instance, the success of many plant invaders is linked to their higher values (i.e., relative to native species) in traits such as maximum height and specific leaf area (Figure 1b), which confer them stronger competitive abilities in hierarchical competition for common resources such as light (14).”

14. Van Kleunen, M., Weber, E., & Fischer, M. (2010). A meta-analysis of trait differences between invasive and non-invasive plant species. *Ecology letters*, 13(2), 235-245.

Line 177: "niche filtering" As in environmental filtering resulting in trait clustering? Perhaps some parentheses here clarifying would help. Sorry, but as you probably know this field is full of redundant jargon!

Response: Yes, by “niche filtering” we were referring precisely to environmental filtering, which drives trait clustering. In our revision of the manuscript, we have replaced the term “niche filtering” with “environmental filtering” as the latter is likely to be more familiar and accessible to general readers.

Line 301: How are the absolute and hierarchical measures calculated at the assemblage level? Above they are described as being calculated between the invader and each native species. Was an average taken to get an assemblage level value? Apologies if I missed this.

Response: Precisely, an average was taken at the assemblage level, as the referee suggests. We have stated this:

“...we modelled Invader Abundance as a function of the average trait difference between *S. invicta* and the assemblage of ant species in each plot.”

For the referee’s interest, this approach (modelling invasion success as a function of the average trait difference between the invader and the species in each assemblage) has previously been used as a recommended approach in multiple trait-based studies of plant invasions (e.g., Carboni et al., 2016; also reviewed in Gallien & Carboni, 2017).

Carboni, M., Münkemüller, T., Lavergne, S., Choler, P., Borgy, B., Violle, C., ... & Thuiller, W. (2016). What it takes to invade grassland ecosystems: Traits, introduction history and filtering processes. *Ecology letters*, 19(3), 219-229.

Gallien, L., & Carboni, M. (2017). The community ecology of invasive species: where are we and what's next?. *Ecography*, 40(2), 335-352.

Figure 3: The y axis labels are misleading as these data are not the size or value of the spaces/traits, but are the dispersion in them. Perhaps relabel to make it clearer. This is clear in the legend but should also be obvious from the label.

Response: To improve the clarity of Figure 3, we have since revised the labels of the x- and y-axes, as follows:

x-axis labels: from “Uninvaded” and “Invaded” to “Uninvaded plots” and “Invaded plots”

y-axis labels: The phrase “Dispersion in” is now included before the name of each trait variable, so as to clarify the precise aspect of trait diversity that is being shown, for instance, “Dispersion in Head Width”.

Line 384: What advantage does a wider head per se confer? This is just an example, but I think more detail needs to be given as to the benefit that these different traits may offer in the grassland.

We have since elaborated on the potential advantages that a wider head may confer to an ant in general, as well as in the specific context of an ant invasion in a grassland setting:

“As the heads of ant workers contain the key musculature involved in the tearing and manipulation of food items (39), wider heads may relate to more efficient resource exploitation. Previous work has shown that *S. invicta* can significantly deplete arthropod prey populations (20); furthermore, exploitative competition for resources among ant species can be especially intense in environments with low habitat complexity (40), such as grasslands.”

(Our next response addresses the following comments by the referee, which pertain to the same issue; the comments have been grouped here for clarity)

I have only one major issue which is the use of the hierarchical difference metric. I have discussed this in detail below. In short, I am sceptical as to whether is telling the authors what they think it is, and whether its potential behaviour in different community contexts (assuming the invader is always “top” of the hierarchy for each trait) may lead to spurious results. The authors may be able to bat away my confusion - great! - and better explain how this metric works in the manuscript. Otherwise, some more thought may be needed in terms of trying to distinguish between these different absolute and hierarchical mechanisms. The absolute difference metric is really intuitive and makes a lot of sense.

Figure 1: Maybe this gets cleared up later, but I'm reading it here as it is referred to in the intro. Figure 1b assumes that the invader has the most competitive trait and that it is always larger than the other traits available. Am I interpreting that correctly? Does the analysis not pick up the effect when the invader has a small value (i.e. in a case when small bodies may do better). Or am I overinterpreting a conceptual figure? I guess this is a larger point about what the trait hierarchy actually is, and whether the assumption running throughout is that the invader *does* have the most competitive trait(s).

Line 295: OK, so the assumption is that the invader has the most competitive trait? I think this needs to be explained in more detail somewhere.

Line 307-310: This is potentially a real issue and makes interpreting these different effects difficult. It also means that using these measures only works when the invader or target species does not have the largest or smallest trait value. Is this desirable behaviour? I am not sure what

the hierarchical measure is supposed to be telling us, and I'm not convinced that it is behaving in the way the authors think it is - especially when put in an assemblage context.

Response: We appreciate the referee's genuine and keen interest in the approach used in our work. We agree that the hierarchical difference ("directional") trait metric, which is meant to detect trait-mediated differences in species' relative competitive abilities (e.g., for the same resource), may not be as intuitively understood as the absolute dissimilarity ("directionless") trait metric, which is meant to detect trait-mediated differences in species use of different resources. Nonetheless, we have confidence in our use of this approach owing to the long and extensive application of the hierarchical difference metric in the literature, such as in early work on hierarchical competition in experimental plant assemblages (e.g., Keddy, 1989; Goldberg & Landa, 1991); more recent trait-based work attributing differences in species' relative competitive abilities to their hierarchical differences in trait values (e.g., Lasky et al., 2014; Kraft et al., 2015; Funk & Wolf, 2016; Kunstler et al., 2016); and even in invasion contexts, where invasion success (e.g., the abundance of an invasive plant) was modelled as a function of the hierarchical difference between its trait value and those of native species (Van Kleunen et al., 2010; Carboni et al., 2016; Gallien & Carboni, 2017).

With regards to the referee's specific query about Figure 1b, we kindly wish to clarify that in testing whether a difference in trait-mediated competitive abilities drives invasion success, our modelling of invasion success against the hierarchical difference metric ($HD = T_{invader} - T_{resident}$) does not make any implicit *a priori* assumption that the invader necessarily has the 'strongest' value for the trait or is at the 'top' of the hierarchy. Indeed, the referee is right in pointing out that the smallest trait value (e.g., smallest body size) may very well be the most 'competitive' trait in the hierarchy. In this case, if the invader does have the smallest body size, and this confers it a competitive advantage over resident species, one would expect invasion success to be negatively correlated with HD (i.e., in Figure 1b, invasion success would decrease linearly with increasing HD and display a slope of -1, instead of increasing linearly with increasing HD and displaying a slope of 1). However, as we did not wish to confuse readers with this alternative example (where smaller trait values are more competitive), we only illustrated the more intuitive scenario (i.e., the case where larger trait values are more competitive) in Figure 1b. We chose to do this because we borrowed the hierarchical difference approach from plant studies, which have frequently shown that larger values in traits such as height and specific leaf area confer stronger competitive abilities (e.g., Van Kleunen et al., 2010; Kunstler et al., 2016). Nonetheless, based on the referee's useful feedback, we recognize the need to provide some clarification about Figure 1b. Thus, we have added the following statement as a note in the caption for Figure 1.

"Note: Fig. 1b shows the expected relationship when larger trait values confer stronger competitive abilities in a trait hierarchy; if smaller trait values confer stronger competitive abilities instead, the inverse relationship is expected."

We wish to clarify that whilst our illustration of the hierarchical difference concept in Figure 1b only showed one 'version' of a trait hierarchy (i.e., where larger values are more competitive),

our analyses allowed for the detection of competitive advantages driven by small as well as larger trait values. Our study aim was to distinguish between the effects of environmental filtering, limiting-similarity competition, and hierarchical competition in driving invasion success. In line with this, we conducted model selection by asking which trait models better explained invasion success – those that included trait differences measured as Absolute Dissimilarities (which would reflect either environmental filtering or limiting-similarity competition) or those that included trait differences measured as Hierarchical Differences (which would reflect hierarchical competition). When comparing models, we did not take into account the specific *direction* of the effect of AD/HD on Invasion Success – only the overall *quality* of the model (i.e., based on AIC). It was only after we selected the best model for particular trait that we inspected the direction of the relationship, to infer exactly which mechanism explained that trait's effects – environmental filtering (Invasion Success is best predicted by AD and decreases with increasing AD), limiting-similarity competition (Invasion Success is best predicted by AD and increases with increasing AD), or hierarchical competition (Invasion Success is best predicted by HD; it can increase or decrease with HD depending on whether larger or smaller trait values are more competitive, respectively).

It is true that the approach of calculating trait differences as AD and HD does not work well when the focal species has the largest value for the trait among all species, because then AD and HD will be completely correlated. Nonetheless, this is the current state of the art (please see papers cited above published in Nature, PNAS, and Ecology Letters), and unfortunately we are unable to find a workaround for this presently. Nonetheless, it is our hope that our findings – from exploring ant invasions with trait-based approaches that have thus far been overwhelmingly applied to plant studies – can be published so that these interesting issues can be more broadly discussed and explored in the ecological literature.

Carboni, M., Münkemüller, T., Lavergne, S., Choler, P., Borgy, B., Violle, C., ... & Thuiller, W. (2016). What it takes to invade grassland ecosystems: Traits, introduction history and filtering processes. *Ecology Letters*, 19(3), 219-229.

Gallien, L., & Carboni, M. (2017). The community ecology of invasive species: where are we and what's next?. *Ecography*, 40(2), 335-352.

Goldberg, D. E., & Landa, K. (1991). Competitive effect and response: hierarchies and correlated traits in the early stages of competition. *The Journal of Ecology*, 1013-1030.

Funk, J. L., & Wolf, A. A. (2016). Testing the trait-based community framework: Do functional traits predict competitive outcomes?. *Ecology*, 97(9), 2206-2211.

Keddy, P. A., & Shipley, B. (1989). Competitive hierarchies in herbaceous plant communities. *Oikos*, 234-241.

Kraft, N. J., Godoy, O., & Levine, J. M. (2015). Plant functional traits and the multidimensional nature of species coexistence. *Proceedings of the National Academy of Sciences*, 112(3), 797-802.

Kunstler, G., Falster, D., Coomes, D. A., Hui, F., Kooyman, R. M., Laughlin, D. C., ... & Westoby, M. (2016). Plant functional traits have globally consistent effects on competition. *Nature*, 529(7585), 204-207.

Lasky, J. R., Uriarte, M., Boukili, V. K., & Chazdon, R. L. (2014). Trait-mediated assembly processes predict successional changes in community diversity of tropical forests. *Proceedings of the National Academy of Sciences*, 111(15), 5616-5621.

Van Kleunen, M., Weber, E., & Fischer, M. (2010). A meta-analysis of trait differences between invasive and non-invasive plant species. *Ecology letters*, 13(2), 235-245.

Appendix B

Associate Editor

Both reviewers are happy with the changes made by the authors in this revision. There are a few more editorial suggestions by both. However, having made these adjustments the reviewers and I suggest the manuscript should be published in Proc Roy Soc. Well done.

Response: We thank the editor and both referees for their positive assessment of our manuscript, as well as for the opportunity to further improve the presentation of the text. We have carefully revised the manuscript according to the suggestions of both reviewers, and detail our responses to their comments below.

Referee: 3

This is an interesting study and a useful addition to the literature.

Response: We thank the referee for their positive assessment of our manuscript.

I do however have a small number of points that should be addressed:

Methods, page 5, lines 198-202

The authors state that the study took place in “two reserves of open grassland”. The composition of the grassland should be described. Whether the grasslands in question are composed mainly/exclusively of native or exotic grasses will clearly influence how native and invasive ant species interact with them. Therefore, although I assume that grassland in a reserve should be fairly intact ecologically, this needs to be described and defined in more detail. If, conversely, these grasslands are mainly exotic, then this might not be the most suitable environment in which to define ants as invasive or native.

Response: Thank you for this suggestion. We have provided details of the native grass species that dominates this grassland. The revised sentence is as follows.

“The study locations were two reserves of open grassland in Hong Kong dominated by the native grass species *Arundinella setosa*.”

Discussion, page 9, lines 382-383

Because it follows a statement on head width, it seems as if reference (20) infers that arthropods were depleted due to this character trait in fire ants, but (20) is in fact about the impact of polygyne fire ants on native ants and arthropods due to the abundance of worker fire ants produced by colonies. This needs to be worded more clearly.

Response: Thank you for this suggestion. We have reordered the sentences to avoid this potential confusion.

“As the heads of ant workers contain the key musculature involved in the tearing and manipulation of food items (39), *S. invicta*’s relatively wider head may have allowed it to exploit trophic resources more efficiently. Notably, exploitative competition for resources among ant species can be especially intense in environments with low habitat complexity (40), such as grasslands. The relatively higher trophic position of *S. invicta* was consistent with previous observations of this species significantly depleting arthropod prey populations during invasion (21).”

Table S1

The table should include an extra column indicating whether each of the ant species listed is native or exotic/invasive.

Response: Thank you for this useful suggestion. We have added this information to Table S1.

A point that arises on inspection of the other species present is that *S. invicta* is shown by the study to be dominant not only against native species, but also against other species that are invasive elsewhere. This is not highlighted in the text but should be, because it sheds light not only on interactions between native and invasive species, but also between invasive species. The authors should consider (1) inclusion of other exotic/invasive species in samples as an explanatory variable to see whether there is an interaction, or (2) analysing the impact of *S. invicta* on native and invasive species separately, to further test the robustness of their hypotheses.

Response: Thank you for this suggestion. As now reflected in Table S1, besides the invader *S. invicta*, the majority of the other species (i.e., 15 of 16 species) are in fact native to Hong Kong based on data from the Global Ant Biodiversity Informatics database (Guénard et al., 2017). A few species that are native to our study system such as *Monomorium floricola* and *Tapinoma melanocephalum* do have extensive non-native distributions; we assume these are the “invasive” species that the referee mentions in their comment. However, such species are more appropriately considered to be “tramp species” for their associations with anthropogenic disturbance and the lack of evidence thus far that they impact biodiversity and ecosystems in their non-native ranges (i.e., in contrast to truly “invasive” species such as *S. invicta*, for which impacts are widely documented) (Passera, 1994; McGlynn, 1999; Wetterer, 2009;2010). We agree that it would be interesting to investigate the interactions between the truly invasive species in this system (*S. invicta*) and several of those native species that are tramps elsewhere in the world. Nonetheless, we feel that such an investigation would constitute a different ecological question that would entail a significant expansion of the scope, aims, methods and analyses used presently. Hence, we respectfully propose not to include additional analyses in the present manuscript, given that it already presents a large amount of detail (e.g., in terms of hypotheses and traits) and is rather lengthy. Incidentally, we are undertaking another study that investigates the interactions between ant species with different biogeographic histories and ecologies, and believe the referee’s suggestion can be well incorporated into that forthcoming work.

McGlynn, T. P. (1999). The worldwide transfer of ants: geographical distribution and ecological invasions. *Journal of Biogeography*, 26(3), 535-548.

Passera, L. (1994). Characteristics of tramp species. In D. F. Williams, Ed. *Exotic ants: biology, impact, and control of introduced species*. Westview, Boulder, Colorado, USA.

Wetterer, J. K. (2009). Worldwide spread of the ghost ant, *Tapinoma melanocephalum* (hymenoptera: Formicidae). *Myrmecological News*, 12, 23-33.

Wetterer, J. K. (2010). Worldwide spread of the flower ant, *Monomorium floricola* (Hymenoptera: Formicidae). *Myrmecological News*, 13, 19-27.

Referee: 4

I think that disentangle the contribution of limiting-similarity competition and hierarchical competition on species abundances is a relevant question in the field of both community assembly and species' invasions. I think as well that the manuscript is now clearly written and organized, and I am happy to recommend it for publication in the journal after addressing a few minor comments.

Response: We thank the referee for their positive assessment of our manuscript.

Lines 128-130. I would recommend rewriting this sentence to clarify more the message.

Response: We have removed the parentheses and replaced “e.g.” with “such as” so that the sentence reads more clearly. The revised sentence is as follows.

“Here, alien species using niches – such as resources, space, or time – that are dissimilar from those of native species persist in the community by exploiting those unoccupied niches, thereby avoiding intense competition that would otherwise result in their exclusion.”

Lines 161. Clarify which databases you refer (global?) within the parentheses, you can also mention some examples of frequently used datasets in other studies.

Response: We have now clarified that we refer to global trait databases, and cited a relevant paper that reviews global trait databases. The revised sentence is as follows.

“Studies of ecological communities commonly use trait information obtained from secondary sources such as global trait databases (19)...”

19. Gallagher, R. V., Falster, D. S., Maitner, B. S., Salguero-Gómez, R., Vandvik, V., Pearse, W. D., ... & Enquist, B. J. (2020). Open Science principles for accelerating trait-based science across the Tree of Life. *Nature Ecology & Evolution*, 4, 294-303.

Lines 285-290. Did the authors used means or weighted means considering the abundance of other species in the community? It is highly recommendable the use of weighted means when possible. Please clarify if you used this approach.

Response: Thank you for this insightful comment. In our study, we used the mean Absolute Dissimilarity between the invader and other species in the community; this is also termed the “mean distance between each native species and the invader” or “MDNS” by Thuiller et al. (2010) and Gallien et al. (2017) in their respective reviews of trait metrics for investigating invasions (mainly in plants). According to Thuiller et al. (2010), the MDNS “is the most straightforward (measure of trait dissimilarity between the invader and other species) as it is directly derived from Ackerly and Cornwell’s (2007) definition of a niche.” Thuiller et al. (2010) and Gallien et al. (2017) discuss that it is possible to use an MDNS weighted by the relative abundances of other species in the community (i.e., the current suggestion of the referee). However, those authors note that a weighted-MDNS invokes additional assumptions, that is, that “abundant (native) species will be more likely to interact with newly introduced species” (Gallien et al., 2017), and that “native species will contribute to the overall biotic resistance of the community according to their relative abundances” (Thuiller et al., 2010). As our study is an early attempt at extending trait analyses used primarily in plant invasion research to study ant invasions, we decided to refrain from incorporating additional assumptions in our approach. Hence, with respect to using an MDNS versus a weighted-MDNS, we found no reason to make the additional assumption that the invading ant species would necessarily interact or compete most strongly with the most abundant resident ant species as, unlike plants, worker ants are mobile and may encounter a variety of other species regardless of their relative abundances. Nonetheless, we do recognize the value of using a weighted measure of trait dissimilarity and will certainly consider exploring this in future work with more specific hypotheses (e.g., competition relating to density dependence).

Gallien, L., & Carboni, M. (2017). The community ecology of invasive species: where are we and what's next?. *Ecography*, 40(2), 335-352.

Thuiller, W., Gallien, L., Boulangéat, I., De Bello, F., Münkemüller, T., Roquet, C., & Lavergne, S. (2010). Resolving Darwin’s naturalization conundrum: a quest for evidence. *Diversity and distributions*, 16(3), 461-475.

Lines 297-300. Please rewrite and clarify. Did authors build two parallels models one with poisson and other with negative binomial error distribution and then choose between them? What do you mean with observation-level random effect? Please clarify if you have replicates before and which is the group level. If you do not have more than one observation per group, you do not need a hierarchical model.

Response: We have corrected the text to remove the statement mentioning the use of a negative binomial model, which was a typographical error from an earlier version of the manuscript. We have also revised the statement to clarify that we used GLMMs with a Poisson

error distribution, including the main effects as well as an observation-level random effect to address the overdispersion in the response variable. According to Harrison (2014) (cited >700 times), adding an observation-level random effect addresses overdispersion by allowing the model to model “the extra-Poisson variation in the response variable using a random effect with a unique level for every data point.” In our revised statement (pasted below), we cite Harrison (2014) (reference 32) directly so that readers are directed to this useful resource.

“Specifically, we built a full generalized linear mixed effects model (GLMM) with a Poisson error distribution, which included Absolute Dissimilarity, Hierarchical Difference, the environmental covariates Ground Cover and Temperature, and an observation-level random effect to address the overdispersion in Invader Abundance (after 32).”

Harrison, X. A. (2014). Using observation-level random effects to model overdispersion in count data in ecology and evolution. *PeerJ*, 2, e616.

Lines 304. Do you mean “differences in this trait” instead of “differences in trait”.

Response: Yes, thank you for spotting this typographical error. We have since corrected the phrase to “differences in this trait”.

Lines 348-350. Please be more precise about the environmental variables that resulted not significant.

Response: We have revised the paragraph to provide more details on this.

“There was little evidence to suggest that environmental effects alone influenced invasion outcomes. The measures of absolute dissimilarity that best predicted invasion success showed positive (instead of negative) effects on invasion success (Table 1), which was inconsistent with environmental filtering. In addition, the environmental variables of Temperature and Ground Cover were poor predictors of invasion success (Table 1) as well as the functional structure of the assemblages (Supporting information).”

Lines 415-429. Regarding to the last sentence about traits that conferring different competitive abilities in some species and niche partitioning in others; is it possible to misinterpret the role of a trait and associate it with hierarchical competition in case the higher value for the trait is reflecting a different niche but not really competing better than other species because of this trait? I mean, could it happen that ants with higher heads have as well a demographic strategy with higher growth rates, so the reason behind ant success, in fact, is not head size?

Response: Thank you for this enquiry. We are unsure as to whether we fully comprehend the referee’s question, and we apologize if we have misinterpreted this. At present, the state-of-the-art approach for convincingly discerning whether a trait confers a niche difference or a competitive ability difference (also known as ‘fitness difference’) to a given species with respect to other species requires a factorial experiment, where the investigator measures the growth

rate of individuals of the focal species in the presence of conspecifics, heterospecifics, and in the absence of competition. The growth rates are then entered into an appropriate competition model to estimate niche and fitness differences, and subsequent analyses can be used to identify the traits that best correlate with those niche and fitness differences. This powerful approach has been used in a variety of plant studies, such as Kraft et al. (2015) and Perez Ramos et al. (2019). Of course, such a rigorous experimental approach has limited applicability to studies on animal communities, especially observational ones. We recognise this limitation, but nonetheless call for animal studies to at least consider the potential for trait differences between species to reflect differences in niches as well as differences in competitive abilities. This is reflected in a statement in our Discussion, as follows.

“Rigorous tests for these coexistence mechanisms would entail competition experiments measuring demographic parameters such as invasion growth rates (e.g., 45). However, such approaches are not readily transferable to animals (see 15). Still, we call for studies of animal communities to consider a more nuanced view of community assembly, such as by exploring how multiple assembly processes may determine community structure through their distinct effects on separate traits. In particular, the potential for both forms of trait-mediated competition (i.e., limiting-similarity and hierarchical competition) to influence community assembly should not be overlooked. Practically, this means not ruling out the role of competition and biotic filtering in community assembly simply when trait clustering is observed (as argued by 16,17).”

Kraft, N. J., Godoy, O., & Levine, J. M. (2015). Plant functional traits and the multidimensional nature of species coexistence. *Proceedings of the National Academy of Sciences*, 112(3), 797-802.

Pérez-Ramos, I. M., Matías, L., Gómez-Aparicio, L., & Godoy, Ó. (2019). Functional traits and phenotypic plasticity modulate species coexistence across contrasting climatic conditions. *Nature communications*, 10(1), 1-11.

Figure 2. (Regarding the methods behind the figure). Did you check the correlation between CTmax.HD and CTmax.AD?

Response: Thank you for suggesting this. Yes, we did check this, and the two metrics were poorly correlated ($r = 0.183$), suggesting they could be included in the same model (as reported). We have also added a statement in the methods to clarify that checks for multicollinearity were performed prior to running the models.

“Checks for multicollinearity were performed prior to running the GLMMs.”